# IDENTITY BRIDGE: ENABLING IMPLICIT REASONING VIA SHARED LATENT MEMORY

## ABSTRACT

Despite remarkable advances, large language models often fail at compositional reasoning tasks, a phenomenon exemplified by the "curse of two-hop reasoning". The inability of large models to perform implicit reasoning as expected remains an intriguing mystery. In this work, we unexpectedly discover that even a simple, single-layer Emb-MLP model can effectively learn to generalize out-of-distribution (OOD) multi-hop reasoning. Specifically, we demonstrate that transformers can successfully address multi-hop tasks through simple shortcuts rather than implicit reasoning with the help of Identity bridge. Our experiments and theoretical analysis established the learning mechanisms of the Emb-MLP model, which is the excellent approximation of GPT-2 on multi-hop task. We further compare the performance of the Emb-MLP model with GPT-2 across various complexities of two-hop reasoning and identify that smaller initializations or larger weight decay parameters enhance the model's ability to harness implicit reasoning. Finally, we generalize our observations to real-world LLMs, presenting evidence that they implicitly acquire this necessary alignment during pretraining, which underpins their compositional abilities.

## 1 INTRODUCTION

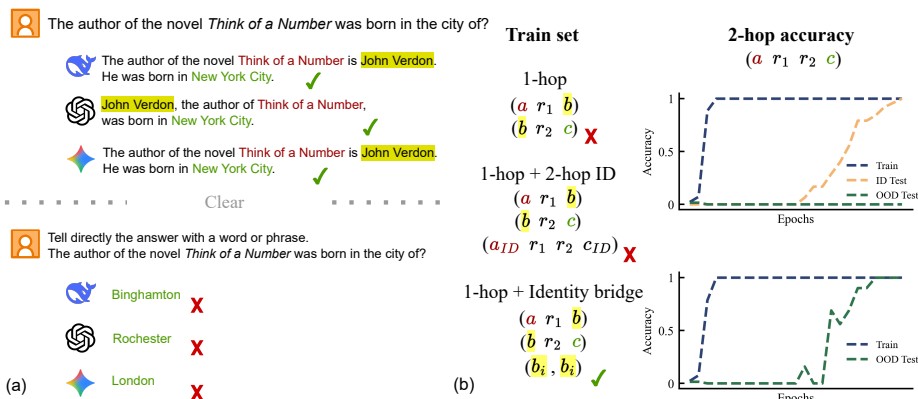

Figure 1: Brief description of the two-hop curse. (a) Large models have difficulty completing two-hop reasoning tasks without CoT assistance. (b) Single-hop tasks and in-distribution two-hop tasks cannot help generalize out-of-distribution two-hop tasks, but identity bridges can.

Large language models (LLMs) achieve strong performance on a wide range of multi-step reasoning tasks, especially when assisted by chain-of-thought (CoT) (Wei et al., 2022; Yao et al., 2023; Kojima et al., 2022). Rather than stopping at the success of CoT, recent works begin to probe the *implicit reasoning* abilities of LLMs, asking what mechanisms and thought patterns actually underlie their reasoning behaviour (Deng et al., 2024; Wang et al., 2024; Bai et al., 2025). A recurring concern in these studies is that models often rely on shortcut reasoning rather than genuine systematic reasoning (Ding et al., 2024; Lin et al., 2025). At the same time, long-standing puzzles in LLM reasoning, such as the reversal curse (Berglund et al., 2024; Allen-Zhu and Li, 2024; 2025; Qi et al., 2023)

and the two-hop curse (Balesni et al., 2025; Yang et al., 2024; Dziri et al., 2023), remain unresolved for modern models, despite being elementary for humans. The latter exposes a core limitation of current training and data: models often fail to compose two single-hop facts ($a \to b$ and $b \to c$ in the training set) into the correct conclusion ($a \to c$ at evaluation) via latent reasoning, a phenomenon we refer to as out-of-distribution (OOD) two-hop reasoning (Wang et al., 2024; Ye et al., 2025). This raises two central questions:

1. Why do vanilla transformers fully fail to generalize on OOD two-hop instances, as observed in Wang et al. (2024), and what minimal modifications to the training data are sufficient to enable such OOD two-hop generalization?

2. When a transformer generalizes, is it executing genuine two-step implicit inference or merely a shortcut based on cross-task memory sharing, despite having enough expressive power for the former (Wang et al., 2025; Qin et al., 2025)?

We revisit this phenomenon following the setup and definition of OOD two-hop reasoning in Wang et al. (2024), where the failure is attributed to the fact that OOD bridge entities never appear as the middle node in any training-time two-hop example. We show that this condition is not necessary for learning OOD two-hop reasoning. Instead, a minimal modification to the training set, which we call the *Identity Bridge*, is sufficient to unlock OOD composition. Concretely, we add a zero-hop task that maps each bridge token to itself. Identity Bridge encourages the model to consistently associate the bridge token $b$ in its two roles: as the output in $a \to b$ and as the input in $b \to c$. Empirically, models that previously failed on OOD two-hop queries begin to compose correctly once Identity Bridge is introduced. This also clarifies a gap between synthetic two-hop settings and pretrained models: pretraining effectively endows models with a latent form of identity bridging (e.g., the ability to restate text), which partially supports composition.

To better understand the underlying mechanism, we theoretically analyze a simplified Emb–MLP model, a one-layer transformer with uniform attention, which has been widely adopted in recent work on transformer reasoning (Yao et al., 2025; Huang et al., 2025; Mahankali et al., 2024; Zhang et al., 2024a). We prove that gradient-descent training in this model induces an implicit nuclear-norm bias toward low-rank, structure-sharing parameters which guarantees OOD generalization from one-hop to two-hop reasoning, even in this one-layer setting. Moreover, our analysis explains why simplified synthetic setups often fail to match the behavior of real-world models (Wang et al., 2024; Ye et al., 2025): they understate the critical role of bridge-token self-consistency.

At the same time, the one-layer solution we uncover for two-hop reasoning is clearly a form of shortcut pattern, rather than the kind of step-by-step implicit reasoning we ultimately desire. Therefore, we further investigated a high-complexity setting with more relationships and empirically demonstrated that the performance degradation of standard GPT-2 is similar to that of the Emb–MLP model, suggesting that Emb–MLP is a good mechanism proxy. However, we noticed the small initialization technique (Zhang et al., 2024b; 2025; Yao et al., 2025) or weight decay is useful to improve OOD generalization in high-complexity regimes. Our analysis links this improvement to tight representation alignment across layers, thereby confirming and extending the explanation in Biran et al. (2024). Finally, we examine pretrained LLMs on real two-hop datasets and find that, even without explicit two-hop supervision, the models tend to associate object $a$ directly with object $c$, suggesting that shortcut-like cross-task memory is prevalent in the latent logic of modern LLMs.

To sum up, our contribution can be summarized as follows.

1. We introduce the Identity Bridge, a minimal modification to the training data that adds a zero-hop task and reliably enables OOD two-hop composition in synthetic settings (Figure 2).

2. We develop a uniform-attention theory (Theorems 1 and 2) showing how Identity Bridge induces cross-task memory sharing that links $a$ to $c$, and why the standard setup without identity supervision fails.

3. We show that in high-complexity regimes, GPT-2 closely matches Emb–MLP, validating Emb–MLP as a good proxy, and that small initialization improves cross-layer alignment and OOD generalization (Figure 6 and Figure 7).

4. We provide evidence that modern LLMs exhibit widespread shortcut-like cross-task memory sharing even without explicit two-hop supervision (Figure 9).

## 2 RELATED WORK

**Implicit reasoning failure on synthetic data** There has been a line of works studying the failure of implicit reasoning on synthetic data, among which the reversal curse and the two-hop inference curse are representative examples.

1. Reversal Curse. The reversal curse (Berglund et al., 2024; Qi et al., 2023) refers to the phenomenon that an auto-regressive LLM that learns "A to B" during training fails to generalize to the reverse direction "B to A". A number of studies have sought to overcome the reversal curse through different strategies, such as extending causal attention to a bidirectional form (Lv et al., 2023), incorporating reversed training samples (Golovneva et al., 2024), applying permutations to semantic units (Guo et al., 2024), or augmenting datasets with reverse-logic instances (Luo et al., 2024). Theoretically, the literature (Zhu et al., 2024) explains the cause of the formation of the reversal curse from a dynamic perspective.

2. Two-Hop Curse. (Allen-Zhu and Li, 2025) studies a phenomenon in which transformers face difficulty in manipulating already learned knowledge. (Press et al., 2022) constructs a two-hop reasoning task on a knowledge graph, demonstrating that transformers can generalize to in-distribution data through long-term training, a phenomenon known as grokking. However, out-of-distribution data cannot be generalized. (Ye et al., 2025) further investigated this phenomenon, demonstrating that in-distribution generalization stems from the presence of bridge entities in the two-hop task in the training set, thereby inducing alignment. However, these works did not propose methods to alleviate the generalization difficulties of OOD.

**Compositionality gap in LLMs** A large body of work has studied the evidence underlying reasoning in LLMs. (Press et al., 2022; Xu et al., 2024) observed a significant gap between the accuracy of single-hop and double-hop tasks in large models, and this gap does not decrease as the model size increases. (Yang et al., 2024) found limited evidence for implicit reasoning in large models by eliminating shortcuts, and (Yu, 2024) found a similar phenomenon in fine-tuning. (Kazemi et al., 2023) also found that the model is more able to utilize popular knowledge rather than lesser-known knowledge during fine-tuning, which affects the model's reasoning performance. These works have demonstrated that it is possible for models to exploit shortcuts instead of golden inference. In order to analyze the possible reasons, (Biran et al., 2024) analyzes the intermediate states in the transformer through circuit analysis and points out that one possible reason is that the first-hop task is completed too late, making the model unable to utilize the first-hop information.

## 3 PRELIMINARIES

### 3.1 2-HOP REASONING TASK SETUP

To accurately investigate the performance of Transformers on OOD 2-hop reasoning tasks, we follow the experimental setups in (Wang et al., 2024; Ye et al., 2025). Specifically, our dataset consists of three distinct components. The first two are used for training, while the third serves exclusively for evaluation:

- **1-hop Facts:** These follow the structure $(a, r_1) \to b$ and $(b, r_2) \to c$. Intuitively, these correspond to real-world facts such as (Alice, mother-of) $\to$ Beth and (Beth, sister-of) $\to$ Carol. Since prior work has demonstrated that In-Distribution (ID) training does not improve OOD 2-hop generalization, we do not distinguish between ID and OOD for these 1-hop facts.

- **Identity Bridge:** These are formatted as $(b) \to b$. This category has been overlooked in previous studies, based on the common assumption that Transformers can trivially align the output $b$ from the first hop (i.e., $(a, r_1) \to b$) with the input $b$ required for the second hop (i.e., $(b, r_2) \to c$). However, we argue that this alignment is non-trivial. We explicitly incorporate this atomic data to facilitate the model's reasoning capabilities.

- **2-hop Queries:** These queries follow the pattern $(a, r_1, r_2) \to c$, corresponding to compositional chains like (Alice, mother-of, sister-of) $\to$ Carol. This subset is strictly reserved for testing and is not seen during training.

**Data complexity.** To provide intuitive insights without loss of generality, our theoretical analysis assumes a single relation type for each hop. For instance, we consider a scenario with only two fixed relations: $r_1$ representing "mother-of" and $r_2$ representing "sister-of". Even in this simplified setting, Transformers fail to generalize from 1-hop facts to 2-hop reasoning without additional intervention.

For the sake of completeness, our experiments extend this setup to consider $C$ distinct relation types for the first hop (where $C > 1$). Specifically, we ensure that the number of bridge entities b associated with each relation variant $r_{1,j_1}, j_1 = 1, \ldots, C$ is balanced. Detailed notation and definitions are available in Appendix A.

### 3.2 MODEL ARCHITECTURE

**Transformer.** We use a standard GPT-2 model. Let $d_{\text{vob}}, d_m, d_k$ denote the vocabulary size, embedding space dimension, and query-key-value projection dimension, respectively. Then, each sequence $x_{1:s}$ is embedded to $\boldsymbol{E}_X$ through embedding matrix $\boldsymbol{E} \in \mathbb{R}^{d_{\text{vob}} \times d_m}$. Then use standard attention and MLP modules, and use residual connections and layer norm between each module. For details on the model implementation, please refer to appendix.

**Embedding-MLP.** For tasks where attention serves only as an information-mixing mechanism, we adopt the Embedding-MLP model (Yao et al., 2025; Huang et al., 2025), which can be viewed as a transformer layer with uniform attention. This formulation allows flexible handling of the input and output vocabularies, denoted by $\mathcal{V}_{\text{in}}$ and $\mathcal{V}_{\text{out}}$. The embedding matrix is $\boldsymbol{E} \in \mathbb{R}^{|\mathcal{V}_{\text{in}}| \times d_m}$, with row $\boldsymbol{E}_x$ corresponding to token $x \in \mathcal{V}_{\text{in}}$, and the projection matrix is $\boldsymbol{W}_{\text{proj}} \in \mathbb{R}^{d_m \times |\mathcal{V}_{\text{out}}|}$. For a sequence $X = (x_1, \ldots, x_s)$, we define:

**Definition 1** (Embedding–MLP (Emb-MLP)). *Given parameters $\boldsymbol{\theta} = (\boldsymbol{E}, \boldsymbol{W}_{\text{proj}})$, the model outputs logits is*

$$f_{\boldsymbol{\theta}}(X) = \left( \sum_{t=1}^{s} \boldsymbol{E}_{x_t} \right) \boldsymbol{W}_{\text{proj}} \in \mathbb{R}^{|\mathcal{V}_{out}|}.$$

### 3.3 PARAMETER INITIALIZATION

For any learnable weight matrix $\boldsymbol{W} \in \mathbb{R}^{d_1 \times d_2}$, with $d_1$ and $d_2$ denoting the input and output dimensions, we initialize each entry from a Gaussian distribution $\boldsymbol{W}_{i,j} \sim \mathcal{N}(0, \sigma^2)$. The standard GPT-2 model take $\sigma = 0.02$. When we use a small initialization, we let $\sigma = d_1^{-\gamma}$, where $\gamma > 0.5$ since related work (Zhang et al., 2024b; Yao et al., 2025) shows that networks initialized in this way often have stronger reasoning capabilities.

## 4 RESULTS

In this section, we construct different construct data sets of different complexity to demonstrate the role of identity bridge which unlocks out-of-distribution composition. To further elucidate the mechanism of the identity bridge, we use embedding-MLP as a simplified model on a dataset with complexity one to prove that the identity bridge, together with the regularization of the gradient descent algorithm, enables the model to share latent space memory, thereby completing two-hop reasoning. On high-complexity data, we use small initialization settings to provide additional alignment effects and achieve generalization.

### 4.1 IDENTITY MATTERS FOR OOD GENERALIZATION

Figure 2 reports test accuracy across models and complexity levels $C$. Identity bridges is effective throughout since it yields non-zero OOD two-hop generalization for all models regardless of whether there are nonlinearities in the model or using small initialization. In particular, at $C = 1$ the identity signal suffices for small-initialized GPT-2, standard GPT-2, and the simplified Emb–MLP to perform well, consistent with the implicit-regularization account developed in Secs. 4.2 and 4.3.

It should be noted that, as complexity $C$ increases, accuracy for standard GPT-2 and Emb–MLP declines simultaneously, indicating that Emb–MLP captures the operative mechanism of the trans-

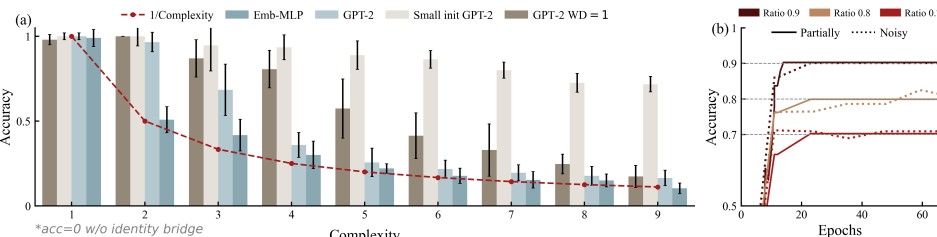

Figure 2: Left: Test accuracy of GPT-2 and Emb-MLP models on datasets with varying complexities. The accuracy of both the standard GPT-2 and Emb-MLP models remains consistent across different complexity levels. Regularization techniques, such as weight decay or small initialization, mitigate the decline in the model's reasoning ability, with small initialization performing better. Right: The effect of retaining a certain proportion of identity bridge, while masking or adding label noise to the remaining. The model's accuracy closely aligns with the proportion of retained.

former and that gradient-descent implicit bias alone is insufficient for strong OOD composition at higher complexity. In contrast, the small-initialized GPT-2 degrades more gracefully, suggesting a stronger regularization effect: the initialization-induced bias further constrains the latent geometry and better preserves the bridge–object coupling needed for composition which will be discussed in Sec. 4.4.

### 4.2 Cross-task memory via implicit regularization

We now examine how identity bridge enables composition of one-hop tasks. To make the mechanism explicit, we use the Emb–MLP to solve task with one complexity and analyze the row-wise logit templates encoded by

$$\boldsymbol{W} \;=\; \boldsymbol{E}\,\boldsymbol{W}_{\mathrm{proj}} \;\in\; \mathbb{R}^{|\mathcal{V}_{\mathrm{in}}| \times |\mathcal{V}_{\mathrm{out}}|},$$

where the $i$-th row of $\boldsymbol{W}$ is the (unnormalized) logit vector produced by input token $i$; specifically, $\boldsymbol{W}_{ij}$ is the logit assigned by token $i$ to output token $j$. Let the input and output vocabularies be $\mathcal{V}_{\mathrm{in}} = \mathcal{E}_1 \cup \mathcal{E}_2 \cup \mathcal{R}$ and $\mathcal{V}_{\mathrm{out}} = \mathcal{E}_2 \cup \mathcal{E}_3$, respectively. For Standard GPT-2 model, we directly use logit as a comparison.

In this setup, Fig. 3(a1) shows that relations act primarily as set selectors: $r_1$ boosts logits toward $\mathcal{E}_2$ while suppressing $\mathcal{E}_3$, and $r_2$ does the converse. Hence, the substantive computation of the two hops is carried by the entity rows of $\boldsymbol{W}$. The key question is whether the subject rows for $a_i$ encode a discriminative bias toward the correct tail $c_i$.

With identity supervision, Fig. 3(a1) further indicates that each bridge token $b_i$ is both self-peaked and object-aligned, exhibiting high logit on $b_i$ and on its paired $c_i$. Training on $(a_i, r_1)$ concentrates subject logits on the appropriate bridge slice; under the implicit nuclear-norm regularization induced by gradient-based training, as discussed in Sec. 4.3, the lowest-rank way to satisfy all constraints shares this structure across blocks, effectively transferring the bridge's object-aligned peak to the subject rows. Consequently, the rows associated with $a_i$ inherit a tail-directed bias via their linkage to $b_i$, supplying the cross-task memory needed for composition. Figs. 3(a2) and 3(a3) show that the model then completes both the one-hop task and the two-hop generalization by combining subject entities with relations.

In contrast, without identity, non-label logits within a block tend to equalize, yielding a nearly diagonal-dominant pattern that conveys little information about the correct object $c_i$ for a given subject $a_i$, thereby leading to failure of two-hop reasoning.

To demonstrate that Emb-MLP is a good approximation of the standard transformer model, we visualized the logit of the standard GPT-2 model. The logit structures for single tokens, one-hop data, and two-hop data all show consistency with Emb-MLP, indicating that although there are differences between the Emb-MLP model and the transformer model, the Emb-MLP model reproduces mechanism of the transformer well in this task.

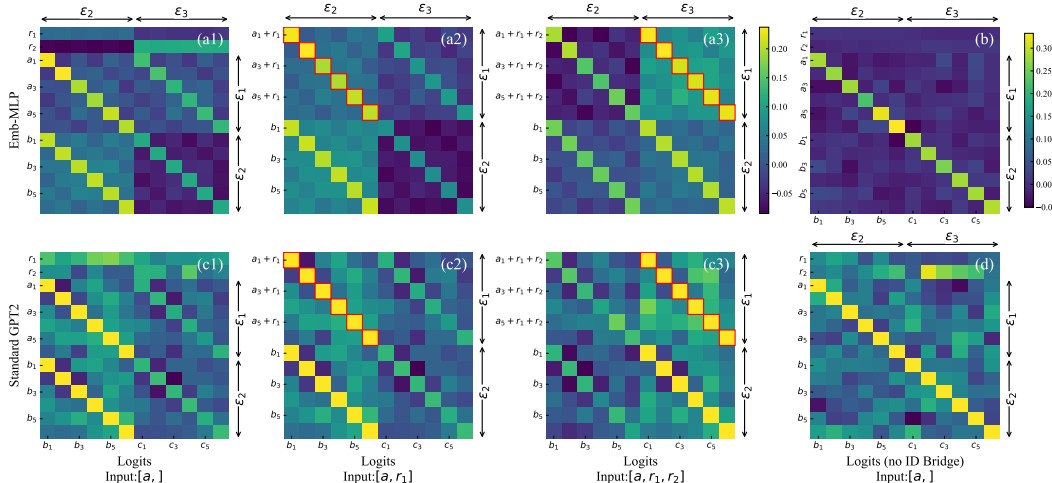

Figure 3: **Row-wise logit templates in Emb–MLP and comparison with Standard GPT-2 at complexity $= 1$ regime.** Panels (a1-a3) and (c1-c3) are trained with identity bridge; panel (b, d) omits it. **(a1)** Base logit matrix: the top two rows correspond to relation tokens $r_1$ and $r_2$; the remaining rows correspond to entity tokens in $\mathcal{E}_1$ and $\mathcal{E}_2$. **(a2)** One-hop input $(a_i, r_1)$, visualized as the row-wise sum of $a_i$ and $r_1$. **(a3)** Two-hop query $(a_i, r_1, r_2)$, visualized as the row-wise sum of $a_i$, $r_1$, and $r_2$. Red boxes indicate the current argmax output token. **(b, d)** Same visualization as in (a1) and (c1) but trained without identity supervision.

## 4.3 UNIFORM-ATTENTION THEORY

We analyze how identity bridge enables two-hop generalization in the Emb-MLP model on the dataset with complexity one, where attention acts only as uniform mixing. Previous experimental evidence has shown that this model contains similar mechanisms to the standard GPT-2 model.

Before presenting the main results, we introduce some necessary concepts and related results. For a labeled example $(X, y)$ with $y \in \mathcal{V}_{\text{out}}$, define the pairwise logit gap and multiclass margin by

$$s_{(X,y),y'} = f_{\boldsymbol{\theta}}(X)_y - f_{\boldsymbol{\theta}}(X)_{y'}, \qquad q(X,y) = \min_{y' \in \mathcal{V}_{\text{out}} \setminus \{y\}} s_{(X,y),y'}.$$

Because Emb-MLP model is positively homogeneous in its parameters, and under standard separability with cross-entropy training, the normalized direction $\boldsymbol{\theta}/\|\boldsymbol{\theta}\|_2$ converges to a KKT point of the margin-maximization program (cf. (Lyu and Li, 2019)):

$$\min_{\boldsymbol{\theta}} \ \tfrac{1}{2}\|\boldsymbol{\theta}\|_2^2 \quad \text{s.t.} \quad s_{(X,y),y'} \geq 1 \ \forall (X,y) \in \mathcal{D}_{\text{train}}, \ \forall y' \in \mathcal{V}_{\text{out}} \setminus \{y\}. \tag{1}$$

In the Emb–MLP model, the logit matrix $\boldsymbol{W}$ can be viewed as a reparameterization of $\boldsymbol{E}$ and $\boldsymbol{W}_{\text{proj}}$, which leads to the following equivalent convex reformulation in $\boldsymbol{W}$ (e.g., (Huang et al., 2025)):

$$\min_{\boldsymbol{W}} \ \tfrac{1}{2}\|\boldsymbol{W}\|_*^2 \quad \text{s.t.} \quad s_{(X,y),y'} \geq 1 \ \forall (X,y) \in \mathcal{D}_{\text{train}}, \ \forall y' \in \mathcal{V}_{\text{out}} \setminus \{y\}. \tag{2}$$

Problem (2) is equivalent to problem (1) based on Lemma 1 in (Huang et al., 2025) whose idea is similar to (Recht et al., 2010). We will briefly introduce the idea behind the equivalence proof; for more details, please refer to (Huang et al., 2025). The objective function of problem (1) for Emb-MLP has the form $\min_{\boldsymbol{E}, \boldsymbol{W}_{\text{proj}}} \tfrac{1}{2} \left( \|\boldsymbol{E}\|_{\text{F}}^2 + \|\boldsymbol{W}_{\text{proj}}\|_{\text{F}}^2 \right)$. Using the connection between the nuclear norm and Frobenius norm that $\|\boldsymbol{W}\|_*^2 = \min_{\{\boldsymbol{E}\boldsymbol{W}_{\text{proj}}=\boldsymbol{W}\}} \tfrac{1}{2} \left( \|\boldsymbol{E}\|_{\text{F}}^2 + \|\boldsymbol{W}_{\text{proj}}\|_{\text{F}}^2 \right)$, we can get the equivalence.

This equivalent convex reformulation brings convenience to analysis. Because it is convex since the objective function is convex and constraints are affine. Therefore, we use problem (2) to state margin consequences for two-hop generalization

We now state the formal consequences for two-hop generalization.

**Theorem 1** (Positive OOD margin with identity supervision). *Given Assumption 1 and large $N$. Assume training includes zero-hop identity supervision over $\mathcal{E}_2$, and let $\boldsymbol{W}^\star$ (equivalently $\boldsymbol{\theta}^\star$) solve*

*problem (2). Then for every OOD query $X = (a_i, r_1, r_2)$ with label $y = c_i$, the multiclass margin is positive:*

$$q(X, y) > 0.$$

*Hence, the composed mapping $g_2 \circ g_1$ is recovered in the OOD two-hop task.*

**Proof sketch.** We first use the permutation symmetry of the dataset to get a highly structured optimal solution and obtain a closed-form objective function in this structure. By further utilizing symmetry, we proved a set of symmetry conditions (Proposition 2) that an optimal solution must satisfy, further simplifying the problem. Next, using the construction method and proof by contradiction, we strengthened the conditions (Proposition 5 and Corollary 1) that the optimal solution must satisfy. These links shrink the OOD margin check to a one-dimensional inequality that feasibility makes strictly positive, yielding a positive margin on every two-hop query and the success of OOD generalization. See appendix B.3 for details of the proof.

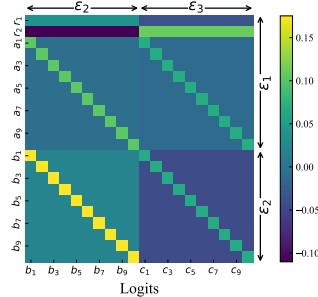

Figure 4: The optimal solution of optimization problem 2 with identity bridge.

**Theorem 2** (Failure without identity supervision). *Given Assumption 1. If identity supervision is omitted, any problem of (2) satisfies, for each OOD query $X = (a_i, r_1, r_2)$ with label $y = c_i$, the multiclass margin is negative:*

$$q(X, y) < 0.$$

*Thus, the composed mapping fails on the OOD two-hop task.*

**Proof sketch.** Without identity supervision, the training constraints are perfectly symmetric inside each block. Because the objective is convex and permutation-invariant, we can average any optimal solution over all within-block permutations and obtain an equally optimal, fully symmetric one. A short KKT check then shows the non-label logits equalize within blocks, so subjects carry no preference toward their true objects. When we test a held-out two-hop composition, the signal from the subject does not point to the correct tail and the margin becomes negative. Hence the margins are negative for all OOD two-hop queries and composition fails. See appendix B.4 for details of the proof.

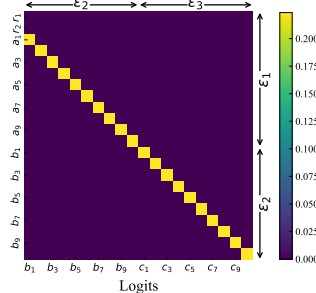

Figure 5: The optimal solution of optimization problem 2 without identity bridge.

### 4.4 HIGH-COMPLEXITY REGIMES: STRENGTHENING REGULARIZATION AND ALIGNMENT

As dataset complexity increases with larger bridge vocabulary and more relation slices, the implicit regularization is no longer sufficient to solve two-hop reasoning task since relying solely on shared latent memory makes the model unable to distinguish the information of the object carried by the subject. We therefore strengthen regularization either by small initialization or by weight decay. Both interventions substantially recover OOD performance.

Figures 6 visualize the geometry of hidden states across layers using the polar alignment plots. As this figure illustrated ,models with small initialization or with weight decay exhibit tighter alignment between $(e_1, r_1)$ and the corresponding bridge $e_2$ than the standard GPT-2 trained with the same dataset.

Furthermore, as Fig. 7 indicates, the emergence of shared latent memory of $a_i$ to $c_i$ precedes the growth of generalization ability. With the alignment of hidden states improves during training, the test accuracy rises in step with this alignment trend. In other words, when the representations for one-hop data and the target bridge collapse into the same subspace, the second hop can reliably latch onto the correct features and composition succeeds.

Mechanistically, both small initialization and weight decay alleviates the pain of the model performing two-hop reasoning only through shared memory. Under higher complexities, We need to further use the information of the bridge entity to enable the model to correctly identify the corresponding object. The alignment of the hidden state allows the model to more directly use the second-hop data to restore positive OOD margins and two-hop generalization.

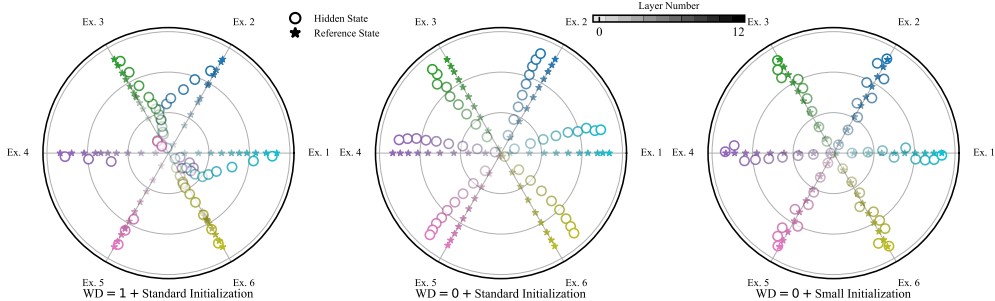

Figure 6: T-SNE visualization of hidden states with respect to one-hop data and corresponding bridge entity. Six samples are selected for each setting, where the star represents the hidden state of the bridge entity and the circle represents the hidden state of the first hop data $(e_1, r_i)$ at position $r_i$. Colors from light to dark represent hidden states from shallow to deep.

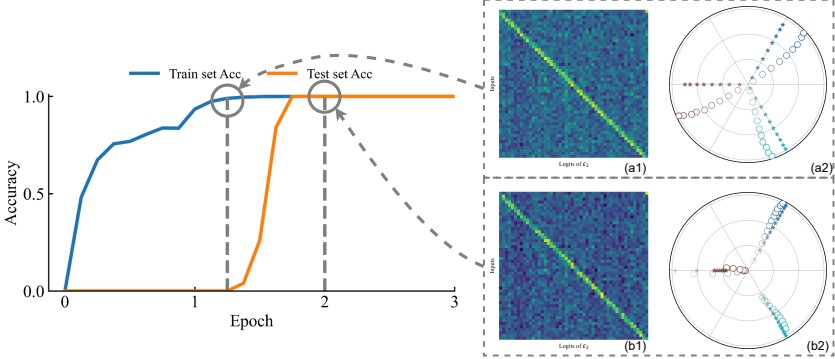

Figure 7: Alignment analysis along the training steps for small initialized GPT-2 with complexity= 2. The two images correspond to when the memorization on 1-hop ends but OOD generalization ability is missing, and the moment when OOD generalization is completed. Figure (a1) and (b1) shows the relationship between the input $a_i$ and the output logits of $c_i$, similarly to Fig. 3. Figure (a2) and (b2) is the T-SNE visualization of hidden states like Fig. 6.

### 4.5 EXTENSION TO 3-HOP REASONING

Although our primary analysis focuses on 2-hop reasoning, the same framework naturally extends to 3-hop reasoning with the single-layer Emb-MLP model. For simplicity, we consider the following chain of 3-hop reasoning:

$$a \xrightarrow{r_1} b \xrightarrow{r_2} c \xrightarrow{r_3} d$$

Based on our theoretical findings, the Emb-MLP model can learn 3-hop reasoning only if it successfully identifies the relationships from $a$ to $d$, which it can extract via the critical token $r_3$. In Fig. 8, we present four different training setups: (a) only 1-hop pairs $(a, r_1, b), (b, r_2, c), (c, r_3, d)$; (b) 1-hop pairs with identity bridges $b \rightarrow b, c \rightarrow c$; (c) 1-hop pairs combined with 2-hop pairs $(a, r_1, r_2, c), (b, r_2, r_3, d)$; and (d) a combination of 1-hop, 2-hop pairs, and identity bridges.

In Fig. 8(a) and (b), the model successfully captures the first and second hop patterns but fails to establish the 3-hop relationship. Training with 2-hop data, however, strengthens the logits from $a \rightarrow c$ and from $b \rightarrow d$, which helps the model successfully complete the $a \rightarrow d$ connection, as shown in Fig. 8(c) and (d).

**Connection to the 'Reversal Curse'.** Some studies have observed that autoregressive large models fail to recognize the reversal of relations, such as not understanding that $a = b$ implies $b = a$. Several works have proposed transformer improvements to address this issue. In Fig. 8(d), we observe that when the output is $c$, there is a clear activation on the diagonal of the group $\mathcal{E}_3 \rightarrow \mathcal{E}_2$ where the output is $b$. This indicates that with the help of the identity bridge, the model can establish

connections such as $b \rightarrow b, c \rightarrow c, b \rightarrow c$, thereby facilitating the model's awareness of the relationship from $c$ to $b$. This suggests that the identity bridge may also offer a potential solution to the Reversal Curse problem.

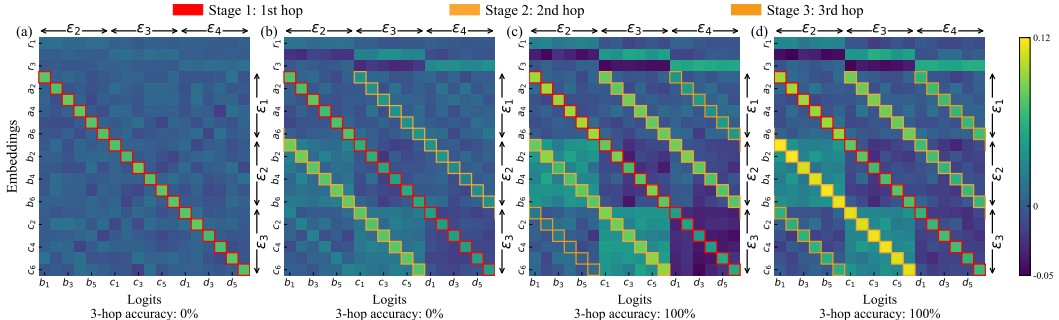

Figure 8: Row-wise logit templates in 3-hop reasoning task $(a, r_1, r_2, r_3, d)$ of Emb–MLP model in different situations: (a) train on 1-hop (b) train on 1-hop and identity bridge (c) train on 1-hop and 2-hop (d) train on 1-hop, 2-hop and identity bridge. 2-hop and identity bridge help the model to construct the relation between $a$ to $d$ highlighted by 'Stage 3'.

## 4.6 REAL TASK ANALYSIS

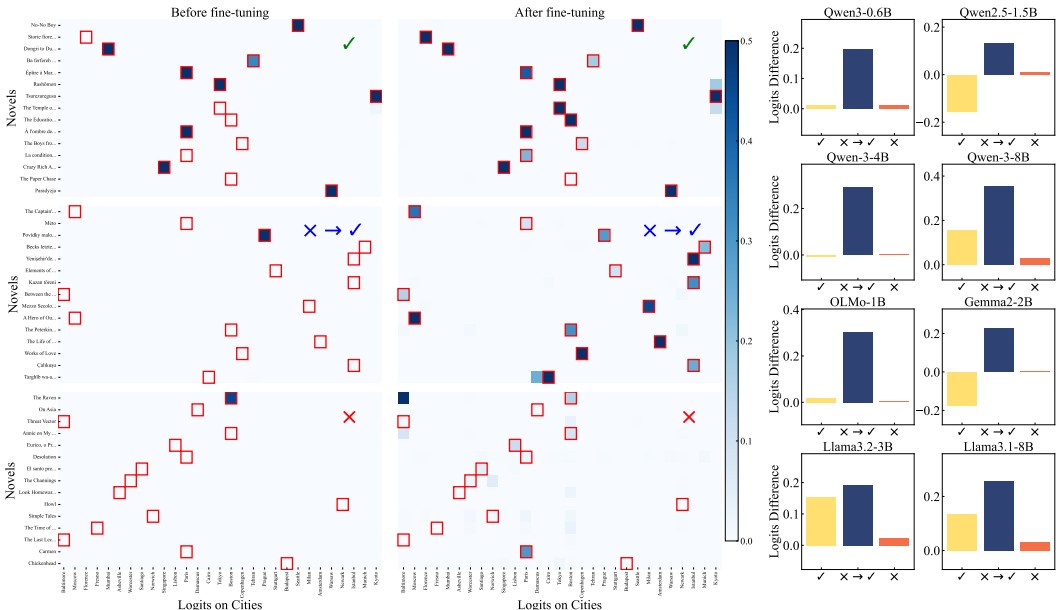

Figure 9: The probability of the model outputting the alternative city token when using the prompt corresponding to $(e_1, r_2)$ before and after fine-tuning on datasets. The red box indicates the answer to the corresponding two-hop reasoning data. The bar chart shows the change in the probability of the corresponding two-hop answer when using the prompt $(e_1, r_2)$ for different datasets.

In this section, we consider the mechanism of two-hop reasoning in real large models. As pointed out in (Press et al., 2022), despite the existence of a combinatorial gap, real large models still have two-hop reasoning capabilities. We finetune pretrained models on TWOHOPFACT dataset which was introduced by (Yang et al., 2024) to verifying our theoretical results. We filtered the data based on the bridge entity to ensure the OOD characteristics of the test data. Although fine-tuning can improve the two-hop reasoning ability of the model to a certain extent, the accuracy improvement brought by the identity mapping is not significant. However, we can still verify the theoretical results on a large model.

For large models, since the parameters have been fully pre-trained, the alignment phenomenon is unlikely to be observed from the hidden state. The model should basically rely on the implicit regularization induced by the gradient descent algorithm to complete the task. In order to observe this mechanism, we extract the following three datasets based on the two-hop results for analysis: $\mathcal{D}_{correct}$ consists of data with correct two-hop reasoning whether fine-tuning or not, $\mathcal{D}_{partial}$ contains the data that are wrong in the first two hops of fine-tuning but correct after fine-tuning, and $\mathcal{D}_{incorrect}$ contains the data of fine-tuning the errors of the two-hop task.

As Fig. 9 shown, after training on the single-hop task, even without seeing the corresponding two-hop data, for the correct data after training, when we use prompts such as $(e_1, r_2)$, such as "the novel was born in the city of" but omit the "author" relation, the model still establishes a strong correlation between the subject and the object, suggesting that the model actually implicitly establishes the identity bridge during the pre-training process which utilizes similar implicit regularization. This is corroborated by our simplified analysis of repetitive tasks using intermediate checkpoints in OLMo-2 (OLMo et al., 2024). For specific details, please refer to Sec. C.3.

We calculated the probability change of labels corresponding to two-hop data using this type of prompt. The results from different models consistently show that completing two-hop reasoning depends on improving the probability of the subject to the object. For the data that still got it wrong after fine-tuning, we found that the probability of the corresponding object was slightly improved. The experiments on other tasks such as MuSiQue dataset (Trivedi et al., 2022) could be referred at Sec. C.4.

## 5 DISCUSSIONS

**Conclusion.** In this work, we revisit the two-hop curse through the lens of implicit reasoning. We show that a minimal modification to the training data, the Identity Bridge, is sufficient to unlock OOD two-hop composition. Through a rigorous analysis of a simplified Emb–MLP model, we prove that identity bridge provably improves OOD two-hop generalization, while its absence inevitably leads to failure.

Empirically, we find that Emb–MLP closely approximates the behavior of standard GPT-2, indicating that standard transformers also tend to solve two-hop tasks via shortcut rather than genuine step-by-step reasoning. To move toward more faithful implicit reasoning, we investigate additional regularization methods, such as small initialization and weight decay, and show that they promote tighter alignment of hidden representations across layers and improve OOD generalization in high-complexity regimes. Finally, experiments on real language tasks suggest that pretraining already endows modern LLMs with implicit identity-bridge-like capabilities, indicating that synthetic data must carefully match the structural properties of real-world tasks when approximating the reasoning behaviour of large language models.

**Limitations.** Our theoretical analysis is conducted within a simplified Emb-MLP model with a uniform attention mechanism and does not explicitly capture the full attention dynamics of transformers. Furthermore, it focuses on two-hop reasoning in synthetic environments. Nevertheless, experiments show that this model tracks GPT-2 well across various complexity levels and reproduces its success and failure patterns with and without identity supervision.

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

# A  NOTATIONS AND DEFINITIONS

We adopt the following notation conventions. All tokens in our synthetic tasks are encoded as positive integers. Calligraphic capital letters (e.g., $\mathcal{E}, \mathcal{R}$) denote sets of tokens, roman capital letters (e.g., $N, C$) denote scalar parameters such as dataset size and complexity, and lower-case letters denote individual tokens. The entity set $\mathcal{E}$ is partitioned into subject, bridge, and object subsets $\mathcal{E}_1, \mathcal{E}_2, \mathcal{E}_3$, with typical elements $e_{1,\cdot} \in \mathcal{E}_1$, $e_{2,\cdot} \in \mathcal{E}_2$, and $e_{3,\cdot} \in \mathcal{E}_3$. The relation set $\mathcal{R}$ is split into first- and second-hop families $\mathcal{R}_1$ and $\mathcal{R}_2$, with elements $r_{1,\cdot} \in \mathcal{R}_1$ and $r_{2,\cdot} \in \mathcal{R}_2$. For any set $S$, we write $|S|$ or $\#S$ for its cardinality.

## A.1  TWO-HOP REASONING TASK SETUP

To study the mechanism behind compositionality gap, we introduce the synthetic dataset in which all tokens are positive integers partitioned into a disjoint set of entities ($\mathcal{E}$) and relations ($\mathcal{R}$). Entities are split into subject, bridge, and object subsets:

$$\mathcal{E} = \mathcal{E}_1 \cup \mathcal{E}_2 \cup \mathcal{E}_3, \quad \text{with} \quad \mathcal{E}_i \cap \mathcal{E}_j = \emptyset \ (i \neq j).$$

Relations are divided into two disjoint families for the two hops:

$$\mathcal{R} = \mathcal{R}_1 \cup \mathcal{R}_2, \quad \text{with} \quad \mathcal{R}_1 \cap \mathcal{R}_2 = \emptyset.$$

**One-hop tasks.** We instantiate two one-hop tasks. For the first hop, we first partition the bridge entities $\mathcal{E}_2$ according to complexity determined by the number of relations in $\mathcal{R}_1$:

$$\mathcal{E}_2 = \bigcup_{j_1=1}^{|\mathcal{R}_1|} \mathcal{E}_{2,j_1}.$$

Then define a deterministic (or sampling) map

$$g_1 : \mathcal{E}_1 \times \mathcal{R}_1 \to \mathcal{E}_2 \quad \text{such that} \quad g_1(e_1, r_{1,j_1}) \in \mathcal{E}_{2,j_1}.$$

The first-hop triple set is

$$\mathcal{T}_1 = \{ (e_1, r_{1,j_1}, e_2) : e_1 \in \mathcal{E}_1, \ r_{1,j_1} \in \mathcal{R}_1, \ e_2 = g_1(e_1, r_{1,j_1}) \}.$$

This partitioning of $\mathcal{E}_2$ ensures that each $r_{1,j_1} \in \mathcal{R}_1$ only co-occurs with bridge entities from its dedicated slice $\mathcal{E}_{2,j_1}$, reducing spurious shortcuts across relations. Second-hop construction follows analogous principles.

**Two-hop composition.** A two-hop instance composes the two one-hop maps. Given $(e_1, r_{1,j_1}, e_2) \in \mathcal{T}_1$ and $(e_2, r_{2,j_2}, e_3) \in \mathcal{T}_2$, the composed query is $(e_1, r_{1,j_1}, r_{2,j_2})$ with answer

$$(e_1, r_{1,j_1}, r_{2,j_2}) = g_2\big(g_1(e_1, r_{1,j_1}), r_{2,j_2}\big) = e_3.$$

**Identity Bridge.** Unlike prior work, we also include a zero-hop task over bridge entities called identity bridge to establish the connection between two one-hop tasks and shape the model's latent space. For each bridge entity $e_2 \in \mathcal{E}_2$ we add a training pair of the form

$$(e_2) \to e_2,$$

encouraging the model to implement an identity transformation $f(e_2) = e_2$ on the bridge entities. This task is not introduced for its standalone difficulty, but to regularize representations so that the composed mapping $g_2 \circ g_1$ can be more reliably recovered during two-hop generalization.

## A.2  DATASET SETUP

**OOD two-hop reasoning.** After determine the $g_1$ and $g_2$ maps, we can construct training dataset $\mathcal{D}_{\text{train}}$ and test dataset $\mathcal{D}_{\text{test}}$ where $\mathcal{D}_{\text{train}}$ contains all one-hop data and partial two-hop data and $\mathcal{D}_{\text{test}}$ contains only the two-hop data. To investigate OOD composition ability, a two-hop $(e_1, r_1, r_2)$ data is called out-of-distribution if the corresponding bridge entity $e_2$ has never appeared in the two-hop

data of the training set. In this work, unless otherwise specified, the training set is restricted to contain only single-hop data, ensuring that all two-hop data are out-of-distribution.

**Dataset Complexity** By adjusting the configurations of $g_1$, $g_2$, and the number of relations, we can control the complexity of the dataset. The dataset complexity is defined precisely as the number of object entities associated with each subject as follows:

$$\text{Complexity} = \max_{e_1 \in \mathcal{E}_1} \#\{e_3 = (e_1, r_{1,j_1}, r_{2,j_2}) \mid r_{1,j_1} \in \mathcal{R}_1, r_{2,j_2} \in \mathcal{R}_2\}.$$

To construct different complexity data, we instantiate families of datasets with controlled complexity. Fix $N \in \mathbb{N}$ and set $|\mathcal{E}_1| = |\mathcal{E}_3| = N$. Let $C \in \mathbb{N}$ denote the complexity parameter and take

$$|\mathcal{E}_2| = CN, \qquad |\mathcal{R}_1| = C, \qquad |\mathcal{R}_2| = 1.$$

Partition the bridge set evenly as

$$\mathcal{E}_2 = \bigcup_{j_1=1}^{C} \mathcal{E}_{2,j_1}, \qquad \mathcal{E}_{2,j_1} = \{ e_{2,k} : (j_1-1)N + 1 \le k \le j_1 N \}.$$

Write $\mathcal{R}_1 = \{r_{1,1}, \dots, r_{1,C}\}$ and $\mathcal{R}_2 = \{r_2\}$. The hop maps are defined by

$$g_1(e_{1,i_1}, r_{1,j_1}) = e_{2,(j_1-1)N+i_1}, \qquad g_2(e_{2,k}, r_2) = e_{3,\lfloor \frac{k}{N} \rfloor + k \bmod N},$$

so that each $r_{1,j}$ selects the $j$-th bridge slice and $r_2$ collapses the bridge index modulo $N$ onto $\mathcal{E}_3$. When $C = 1$, we construct the structured setting used for analysis:

$$\mathcal{E}_1 = \{a_1, \dots, a_N\}, \quad \mathcal{E}_2 = \{b_1, \dots, b_N\}, \quad \mathcal{E}_3 = \{c_1, \dots, c_N\}, \quad \mathcal{R}_1 = \{r_1\}, \mathcal{R}_2 = \{r_2\},$$

with one-to-one correspondences $g_1(a_{i_1}, r_1) = b_{i_1}$ and $g_2(b_{i_2}, r_2) = c_{i_2}$, hence the composed answer for $(a_{i_1}, r_1, r_2)$ is $c_{i_1}$.

# B   THEORETICAL DETAILS

This appendix completes the proofs of Theorems 1 and 2. We first collect several standard tools and assumptions from the literature that will be used throughout the arguments, and then give the proofs via a detailed analysis of the nuclear-norm program—combining a constructive step with a contradiction argument.

## B.1   PRELIMINARIES FOR KKT CONDITIONS

First we recall the notion of subdifferentials for convex functions and then state the KKT conditions in subdifferential form.

Let $\varphi : \mathbb{R}^d \to (-\infty, +\infty]$ be a proper convex function. The subdifferential of $\varphi$ at $\boldsymbol{x} \in \mathbb{R}^d$ is defined by

$$\partial \varphi(\boldsymbol{x}) := \left\{ \boldsymbol{s} \in \mathbb{R}^d : \varphi(\boldsymbol{y}) \ge \varphi(\boldsymbol{x}) + \langle \boldsymbol{s}, \boldsymbol{y} - \boldsymbol{x} \rangle \quad \forall \boldsymbol{y} \in \mathbb{R}^d \right\}.$$

Consider the following optimization problem (P) for $\boldsymbol{x} \in \mathbb{R}^d$:

$$\begin{aligned} \min \quad & f(\boldsymbol{x}) \\ \text{s.t.} \quad & g_n(\boldsymbol{x}) \ge 0 \quad \forall n \in [N], \\ & h_m(\boldsymbol{x}) = 0 \quad \forall m \in [M], \end{aligned}$$

where $f, g_n, h_m$ are convex and finite-valued on their domains. We say that $\boldsymbol{x} \in \mathbb{R}^d$ is a feasible point of (P) if $\boldsymbol{x}$ satisfies $g_n(\boldsymbol{x}) \ge 0$ for all $n \in [N]$ and $h_m(\boldsymbol{x}) = 0$ for all $m \in [M]$.

**Definition 2** (KKT point). *A feasible point $\boldsymbol{x}$ of (P) is called a KKT point if there exist multipliers $\lambda_1, \dots, \lambda_N \ge 0$ and $\mu_1, \dots, \mu_M \in \mathbb{R}$ such that*

*1. (Subdifferential stationarity) There exist subgradients $\boldsymbol{s}_f \in \partial f(\boldsymbol{x})$, $\boldsymbol{s}_{g_n} \in \partial g_n(\boldsymbol{x})$ for all $n \in [N]$, and $\boldsymbol{s}_{h_m} \in \partial h_m(\boldsymbol{x})$ for all $m \in [M]$ such that*

$$\boldsymbol{s}_f - \sum_{n=1}^{N} \lambda_n \boldsymbol{s}_{g_n} - \sum_{m=1}^{M} \mu_m \boldsymbol{s}_{h_m} = 0.$$

*Equivalently,*

$$0 \in \partial f(\boldsymbol{x}) - \sum_{n=1}^{N} \lambda_n \partial g_n(\boldsymbol{x}) - \sum_{m=1}^{M} \mu_m \partial h_m(\boldsymbol{x}).$$

2. *(Complementary slackness) For all $n \in [N]$,*

$$\lambda_n g_n(\boldsymbol{x}) = 0.$$

In general, global minimizers of (P) do not necessarily satisfy the KKT conditions. However, under suitable regularity assumptions (such as a Slater-type constraint qualification), the KKT conditions become necessary for optimality. To ensure the validity and clarity of the subsequent analysis, we impose the following standard assumptions.

Next we recall a standard constraint qualification for convex optimization, known as Slater's condition, which guarantees that KKT conditions are satisfied at optimal solutions.

**Definition 3** (Slater's condition). *Consider problem (P) above and assume that $f$ and $g_n$ are convex functions on $\mathbb{R}^d$ for all $n \in [N]$, and each $h_m$ is an affine function on $\mathbb{R}^d$ for all $m \in [M]$. We say that Slater's condition holds for (P) if there exists a point $\bar{\boldsymbol{x}} \in \mathbb{R}^d$ such that*

$$g_n(\bar{\boldsymbol{x}}) > 0 \quad \forall n \in [N], \qquad h_m(\bar{\boldsymbol{x}}) = 0 \quad \forall m \in [M].$$

*Such a point $\bar{\boldsymbol{x}}$ is called a strictly feasible point.*

Under Slater's condition, the KKT conditions become necessary (and, in the convex setting, also sufficient) for optimality.

**Proposition 1** (KKT conditions under Slater's condition). *Suppose that (P) is a convex optimization problem in the sense that $f$ and $g_n$ are convex and $h_m$ are affine, and that Slater's condition holds. Let $\boldsymbol{x}^*$ be a global minimizer of (P). Then $\boldsymbol{x}^*$ is a KKT point; that is, there exist multipliers $\lambda_1^*, \ldots, \lambda_N^* \geq 0$ and $\mu_1^*, \ldots, \mu_M^* \in \mathbb{R}$ such that $\boldsymbol{x}^*$ and $(\lambda^*, \mu^*)$ satisfy the subdifferential KKT conditions above. Conversely, any KKT point of (P) is a global minimizer.*

In other words, for convex problems satisfying Slater's condition, the KKT conditions provide an exact characterization of optimal solutions.

### B.2 Auxiliary Results from the Literature

We restate the external lemmas and assumptions needed in our proofs, in formulations specialized to our notation. Proofs are omitted and can be found in the citepd references.

**Lemma 1** (Existence of a restricted form solution to (2), (Huang et al., 2025) Lemma 3). *Suppose $\boldsymbol{W}$ is the solution to the optimization problem (2) with identity task. There exists a solution with parameter $a_1, a_2, b_1, b_2, c_1, c_2, d_1, d_2, e, f, g, h$ such that*

$$\boldsymbol{W}^\intercal = \begin{pmatrix} a_1 \boldsymbol{I}_n + a_2 \boldsymbol{E}_n & b_1 \boldsymbol{I}_n + b_2 \boldsymbol{E}_n & e \boldsymbol{1}_n & f \boldsymbol{1}_n \\ c_1 \boldsymbol{I}_n + c_2 \boldsymbol{E}_n & d_1 \boldsymbol{I}_n + d_2 \boldsymbol{E}_n & g \boldsymbol{1}_n & h \boldsymbol{1}_n \end{pmatrix}. \tag{3}$$

*The parameters follow the following constraints:*

$$\begin{aligned} & a_1, d_1 \geq 1, \\ & a_1 + a_2 + e \geq c_1 + c_2 + g + 1, \\ & d_1 + d_2 + h \geq b_1 + b_2 + f + 1. \end{aligned} \tag{4}$$

*In addition, after introducing the identity mapping, the problem gains additional constraints:*

$$\begin{aligned} & b_1 \geq 1, \\ & b_1 + b_2 \geq d_1 + d_2 + 1. \end{aligned} \tag{5}$$

To analyze the optimal solution of optimization problem 2, we need an explicit formula of the nuclear norm. After block multiplication, $\boldsymbol{W}^\intercal \boldsymbol{W}$ can be written as:

$$\boldsymbol{W}^\intercal \boldsymbol{W} = \begin{pmatrix} C_{A1} \boldsymbol{I}_n + C_{A2} \boldsymbol{E}_n & C_{B1} \boldsymbol{I}_n + C_{B2} \boldsymbol{E}_n \\ C_{B1} \boldsymbol{I}_n + C_{B2} \boldsymbol{E}_n & C_{D1} \boldsymbol{I}_n + C_{D2} \boldsymbol{E}_n \end{pmatrix}, \tag{6}$$

where the coefficients are:

$$C_{A1} = a_1^2 + b_1^2$$
$$C_{A2} = 2a_1a_2 + na_2^2 + 2b_1b_2 + nb_2^2 + e^2 + f^2$$
$$C_{D1} = c_1^2 + d_1^2$$
$$C_{D2} = 2c_1c_2 + nc_2^2 + 2d_1d_2 + nd_2^2 + g^2 + h^2$$
$$C_{B1} = a_1c_1 + b_1d_1$$
$$C_{B2} = a_1c_2 + a_2c_1 + na_2c_2 + b_1d_2 + b_2d_1 + nb_2d_2 + eg + fh.$$

The proof of Lemma 1 is provided in (Huang et al., 2025). Readers may also consult the proof of Lemma 4, as the underlying ideas are closely related. Through direct calculation, we have an explicit expression for the nuclear norm of $\boldsymbol{W}$. We introduce the following notation to simplify the formula.

$$\boldsymbol{u} = \left(a_1 + na_2, b_1 + nb_2, \sqrt{n}e, \sqrt{n}f\right)^{\mathsf{T}}, \quad \boldsymbol{v} = \left(c_1 + nc_2, d_1 + nd_2, \sqrt{n}g, \sqrt{n}h\right)^{\mathsf{T}} \quad (7)$$

and

$$A = \begin{pmatrix} a_1 & b_1 \\ c_1 & d_1 \end{pmatrix}, \quad B = (\boldsymbol{u}, \boldsymbol{v}) \quad (8)$$

**Lemma 2** (Closed-form of $\|\boldsymbol{W}\|_*$ under the restricted form). *Let $\boldsymbol{W}$ be in the restricted form Eq. (3). Then its nuclear norm admits the closed form*

$$\|\boldsymbol{W}\|_* = (n-1)\sqrt{C_{A1} + C_{D1} + 2|a_1d_1 - b_1c_1|}$$
$$+ \sqrt{C_{A1} + nC_{A2} + C_{D1} + nC_{D2} + 2\sqrt{(C_{A1} + nC_{A2})(C_{D1} + nC_{D2}) - (C_{B1} + nC_{B2})^2}}. \quad (9)$$

*In particular, Eq. (9) can be reformulated as*

$$\|\boldsymbol{W}\|_* = (n-1)\|A\|_* + \|B\|_*. \quad (10)$$

*Proof.* Eq. (9) follows directly from the Lemma 6 in (Huang et al., 2025). Therefore, we focus on Eq. (10). For the first term, we expand the expression according to the definition.

$$\sqrt{C_{A1} + C_{D1} + 2|a_1d_1 - b_1c_1|} = \sqrt{a_1^2 + b_1^2 + c_1^2 + d_1^2 + 2|a_1d_1 - b_1c_1|} \quad (11)$$

Then, we check it is just $\|A\|_*$. Let $X \in \mathbb{R}^{n \times 2}$ and $\sigma_1, \sigma$ be singular values, we get $\|X\|_* = \sigma_1 + \sigma_2$. We use the following identity

$$\sigma_1 + \sigma_2 = \sqrt{\sigma_1^2 + \sigma_2^2 + 2\sigma_1\sigma_2}. \quad (12)$$

Moreover, we use

$$\sigma_1^2 + \sigma_2^2 = \|X\|_{\mathrm{F}}^2, \quad \sigma_1\sigma_2 = \sqrt{\det(X^{\mathsf{T}}X)} \quad (13)$$

Substitute $A$ into above equation, we get the result. The proof for the second term is similar. $\square$

To fully characterize the properties of the optimal solution to the optimization problem in (2), the mere existence of a solution is insufficient. To address this limitation, (Huang et al., 2025) introduces the following assumption and establishes the uniqueness of the solution.

**Assumption 1.** *Suppose that $\boldsymbol{W}$ is a solution to optimization problem (2) with (or without) identity mapping, but takes a different form from (3) (or (61)). Then we assume that $\mathbf{1}_{2n}^{\mathsf{T}}\boldsymbol{W}^{\mathsf{T}} \neq \mathbf{0}_{2n+2}^{\mathsf{T}}$.*

Based on Assumption 1, the uniqueness theorem is stated as follows.

**Theorem 3** (Uniqueness, rephrased from (Huang et al., 2025) Theorem 5). *Given Assumption 1, the solution of optimization problem (2) with (or without) identity mapping has restricted form (3) (or (61)).*

## B.3 Proof for Theorem 1

We start with the following Proposition which characterize the optimal solution of optimization problem (2) with restricted form.

**Proposition 2.** *Suppose that $W$ is a optimal solution of optimization problem (2) with restricted form (3). Then, $W$ satisfies $\mathbf{1}_{2n}^\mathsf{T} W^\mathsf{T} = \mathbf{0}_{2n+2}^\mathsf{T}$.*

*Proof.* Using the notation introduced in Eqs. (7) and (8) and the Lemma 2, the optimization problem can be reformulated as

$$\min_{a_i, b_i, c_i, d_i, e, f, g, h} \quad (n-1)\|A\|_* + \|B\|_*$$

$$\begin{aligned}
\text{s.t.} \quad & g_1 : a_1 \geq 1, \\
& g_2 : b_1 \geq 1, \\
& g_3 : d_1 \geq 1, \\
& g_4 : a_1 + a_2 + e \geq c_1 + c_2 + g + 1, \\
& g_5 : b_1 + b_2 \geq d_1 + d_2 + 1, \\
& g_6 : d_1 + d_2 + h \geq b_1 + b_2 + f + 1.
\end{aligned} \tag{14}$$

We assert that the optimal solution to the optimization problem satisfies the KKT conditions. To this end, we only need to verify the convexity and the Slater condition.

**Convexity.** Since the nuclear norm is convex, the $\|A\|_*$ is convex with respect to $a_1, b_1, c_1, d_1$. For the term $\|B\|_*$, it is also convex since it is a composition of convex function and affine transformation. Moreover, the optimization problem is convex since the objective function is convex and the constraints are affine.

**Slater condition.** We can construct a solution which is strictly feasible. Here, we give an example. Let $a_1 = 2, b_1 = 3, c_1 = 0.5, d_1 = 1.5, a_2 = 0, b_2 = 0, c_2 = 0, d_2 = 0, e = 0, g = 0, f = 0, h = 4$. Through direct testing, we can see that this solution is strictly feasible.

Thus, we can use the KKT condition to characterize the optimal solution. Observing that $\|A\|_*$ is independent of $a_2, b_2, c_2, d_2, e, f, g, h$, we consider the stationary condition with respect to them since the symmetry of $B$ with respect to $u$ and $v$ will be helpful. Using chain rule, we get

$$\begin{aligned}
(a_2) &: 0 \in n(\partial_{\boldsymbol{u}} \|B\|_*)_1 - \lambda_4, \quad (c_2) : 0 \in n(\partial_{\boldsymbol{v}} \|B\|_*)_1 + \lambda_4 \\
(b_2) &: 0 \in n(\partial_{\boldsymbol{u}} \|B\|_*)_2 - \lambda_5 + \lambda_6, \quad (d_2) : 0 \in n(\partial_{\boldsymbol{v}} \|B\|_*)_2 + \lambda_5 - \lambda_6 \\
(e) &: 0 \in (\partial_{\boldsymbol{u}} \|B\|_*)_3 - \lambda_4, \quad (g) : 0 \in (\partial_{\boldsymbol{v}} \|B\|_*)_3 + \lambda_4, \\
(f) &: 0 \in (\partial_{\boldsymbol{u}} \|B\|_*)_4 + \lambda_6, \quad (h) : 0 \in (\partial_{\boldsymbol{v}} \|B\|_*)_4 - \lambda_6,
\end{aligned} \tag{15}$$

Utilizing the symmetry, we find that there exist $\boldsymbol{s_u} \in \partial_{\boldsymbol{u}} \|B\|_*$ and $\boldsymbol{s_v} \in \partial_{\boldsymbol{v}} \|B\|_*$ such that

$$\boldsymbol{s_u} + \boldsymbol{s_v} = 0. \tag{16}$$

Next, we proceed by case analysis.

1. $\boldsymbol{u} = \boldsymbol{v} = 0$. In this case, we get $f = h = 0$ by the definition of $\boldsymbol{u}$ and $\boldsymbol{v}$. Then the inequality constraints $g_5$ and $g_6$ will be reformulated as

$$b_1 + b_2 \geq d_1 + d_2 + 1, \quad d_1 + d_2 \geq b_1 + b_2 + 1. \tag{17}$$

It makes a contradiction and implies that $\boldsymbol{u} = \boldsymbol{v} = 0$ is not a solution.

2. $\boldsymbol{u} = 0$ or $\boldsymbol{v} = 0$. We just need to consider the $\boldsymbol{u} = 0$ case and the proof of $\boldsymbol{v} = 0$ case is similar. First, we recall the definition of the subderivative of nuclear norm. Let $X \in \mathbb{R}^{m \times n}$ with singular value decomposition $X = U\Sigma V^\mathsf{T}$, the subderivative of nuclear norm with respect to $X$ is

$$\partial \|X\|_* = \{UV^\mathsf{T} + M : U^\mathsf{T} M = 0, MV = 0, \|M\|_2 \leq 1\} \tag{18}$$

Substitute $B = (0, \boldsymbol{v})$ into the definition, we find

$$\boldsymbol{v}^\mathsf{T} M = 0, \quad M\boldsymbol{e}_2 = 0. \tag{19}$$

It implies that

$$\partial \|B\|_* = \{(\boldsymbol{w}, \boldsymbol{v}) : \boldsymbol{w} \perp \boldsymbol{v}, \|\boldsymbol{w}\| \leq 1\}. \tag{20}$$

Then the condition (16) implies that there exists $\boldsymbol{w}$ such $\boldsymbol{w} + \boldsymbol{v} = 0$. It makes a contradiction since $\boldsymbol{w} \perp \boldsymbol{v}$.

3. $\boldsymbol{u} \neq 0$ and $\boldsymbol{v} \neq 0$. We will further discuss two scenarios in this case.

    (a) $\boldsymbol{u}$ and $\boldsymbol{v}$ are linearly independent. In this case, the subgradient equals the gradient because there are no singularities. By direct computation, we get

$$\nabla_{\boldsymbol{u}}\|B\|_* = \frac{1}{\|B\|_*}\left(\boldsymbol{u} + \frac{\|\boldsymbol{v}\|_2^2\boldsymbol{u} - \langle \boldsymbol{u}, \boldsymbol{v}\rangle \boldsymbol{v}}{\sqrt{\|\boldsymbol{u}\|_2^2\|\boldsymbol{v}\|_2^2 - \langle \boldsymbol{u}, \boldsymbol{v}\rangle^2}}\right)$$

$$\nabla_{\boldsymbol{v}}\|B\|_* = \frac{1}{\|B\|_*}\left(\boldsymbol{v} + \frac{\|\boldsymbol{u}\|_2^2\boldsymbol{v} - \langle \boldsymbol{u}, \boldsymbol{v}\rangle \boldsymbol{u}}{\sqrt{\|\boldsymbol{u}\|_2^2\|\boldsymbol{v}\|_2^2 - \langle \boldsymbol{u}, \boldsymbol{v}\rangle^2}}\right) \tag{21}$$

Thus, the condition (16) implies that

$$\frac{1}{\|B\|_*}\left(1 + \frac{\|\boldsymbol{v}\|_2^2 - \langle \boldsymbol{u}, \boldsymbol{v}\rangle}{\sqrt{\|\boldsymbol{u}\|_2^2\|\boldsymbol{v}\|_2^2 - \langle \boldsymbol{u}, \boldsymbol{v}\rangle^2}}\right)\boldsymbol{u} + \frac{1}{\|B\|_*}\left(1 + \frac{\|\boldsymbol{u}\|_2^2 - \langle \boldsymbol{u}, \boldsymbol{v}\rangle}{\sqrt{\|\boldsymbol{u}\|_2^2\|\boldsymbol{v}\|_2^2 - \langle \boldsymbol{u}, \boldsymbol{v}\rangle^2}}\right)\boldsymbol{v} = 0. \tag{22}$$

Since $\boldsymbol{u}$ and $\boldsymbol{v}$ are linearly independent, the coefficient must be zero. It implies

$$\sqrt{\|\boldsymbol{u}\|_2^2\|\boldsymbol{v}\|_2^2 - \langle \boldsymbol{u}, \boldsymbol{v}\rangle^2} = \langle \boldsymbol{u}, \boldsymbol{v}\rangle - \|\boldsymbol{v}\|^2 \quad \text{and} \quad \sqrt{\|\boldsymbol{u}\|_2^2\|\boldsymbol{v}\|_2^2 - \langle \boldsymbol{u}, \boldsymbol{v}\rangle^2} = \langle \boldsymbol{u}, \boldsymbol{v}\rangle - \|\boldsymbol{u}\|^2 \tag{23}$$

which implies $\|\boldsymbol{u}\|_2^2 = \|\boldsymbol{v}\|_2^2$. Then by Cauchy-Schwarz inequality, we get $|\langle \boldsymbol{u}, \boldsymbol{v}\rangle| \leq \|\boldsymbol{u}\|\|\boldsymbol{v}\|$. However, we find $\langle \boldsymbol{u}, \boldsymbol{v}\rangle \geq \|\boldsymbol{u}\|\|\boldsymbol{v}\|$ by Eq. (23) and $\|\boldsymbol{u}\| = \|\boldsymbol{v}\|$. As a result, we find $\boldsymbol{u}$ is parallel to $\boldsymbol{v}$ which makes a contradiction.

    (b) $\boldsymbol{u}$ and $\boldsymbol{v}$ are linearly dependent. In this case, we prove $\boldsymbol{u} = -\boldsymbol{v}$ which finishes the proof. Let $\boldsymbol{v} = \lambda\boldsymbol{u}$ and the problem turns into whether $\lambda = -1$. By the definition of the subgradient of the nuclear norm, we find

$$\partial\|B\|_* = \left\{\boldsymbol{u}(1, \lambda) + M : \boldsymbol{u}^\mathsf{T} M = 0, M(1, \lambda)^\mathsf{T} = 0, \|M\|_2 \leq 1\right\}. \tag{24}$$

Let $M = (\boldsymbol{w}_1, \boldsymbol{w}_2)$, we will find

$$\boldsymbol{u}^\mathsf{T}\boldsymbol{w}_1 = 0, \quad \boldsymbol{u}^\mathsf{T}\boldsymbol{w}_2 = 0 \tag{25}$$

Based on condition (16), we have

$$\boldsymbol{u}(1 + \lambda) + (\boldsymbol{w}_1 + \boldsymbol{w}_2) = 0. \tag{26}$$

Since $\boldsymbol{u}$ is perpendicular $\boldsymbol{w}_1, \boldsymbol{w}_2$, we get $1 + \lambda = 0$ and finish the proof.

$$\square$$

Using Proposition 2, the nuclear norm minimization problem (2) can be simplified to a more tractable form. The following lemma provides this equivalent formulation.

**Lemma 3.** *The optimization (2) problem is equivalent to the following optimization problem.*

$$
\begin{aligned}
\min_{a_i, b_i, c_i, d_i, e, f, g, h} \quad & (n-1)\sqrt{M_1} + \sqrt{2M_2} \\
\text{s.t.} \quad & a_1, b_1, d_1 \geq 1, \\
& a_1 + a_2 + e \geq c_1 + c_2 + g + 1, \\
& b_1 + b_2 \geq d_1 + d_2 + 1, \\
& d_1 + d_2 + h \geq b_1 + b_2 + f + 1, \\
& a_1 + c_1 = -n(a_2 + c_2), \quad b_1 + d_1 = -n(b_2 + d_2), \\
& e = -g, \quad f = -h,
\end{aligned}
\tag{27}
$$

*where the terms $M_1$ and $M_2$ are defined as*

$$
\begin{aligned}
M_1 &= a_1^2 + b_1^2 + c_1^2 + d_1^2 + 2|a_1 d_1 - b_1 c_1|, \\
M_2 &= (a_1 + na_2)^2 + (b_1 + nb_2)^2 + ne^2 + nf^2.
\end{aligned}
$$

*Proof.* The proof consists of two main steps. First, we establish several key identities among the problem's coefficients that arise from Proposition 2. Second, we substitute these identities into the nuclear norm expression from (9) to derive the simplified objective function.

**Step 1: Deriving Identities from Proposition 2.** The four equality constraints presented in the lemma statement are a direct consequence of the structural properties imposed by Proposition 2. Their derivation involves straightforward algebraic manipulation and is omitted for brevity.

These equalities lead to two crucial identities for the aggregated coefficients. First, we show that $C_{A1} + nC_{A2} = C_{D1} + nC_{D2}$. By expanding the terms, we have:

$$C_{A1} + nC_{A2} = a_1^2 + b_1^2 + n\left(2a_1a_2 + na_2^2 + 2b_1b_2 + nb_2^2 + e^2 + f^2\right)$$
$$= (a_1 + na_2)^2 + (b_1 + nb_2)^2 + ne^2 + nf^2.$$

Using the equality constraints like $a_1 + c_1 = -n(a_2 + c_2)$, the expression above is equivalent to $(c_1 + nc_2)^2 + (d_1 + nd_2)^2 + ng^2 + nh^2$, which is precisely the expansion of $C_{D1} + nC_{D2}$.

Second, we analyze the cross-term $C_{B1} + nC_{B2}$:

$$C_{B1} + nC_{B2} = a_1c_1 + b_1d_1 + n(a_1c_2 + a_2c_1 + na_2c_2 + b_1d_2 + b_2d_1 + nb_2d_2 + eg + fh)$$
$$= (a_1 + na_2)(c_1 + nc_2) + (b_1 + nb_2)(d_1 + nd_2) + neg + nfh.$$

Applying the equality constraints again, this simplifies to:

$$C_{B1} + nC_{B2} = -(a_1 + na_2)^2 - (b_1 + nb_2)^2 - ne^2 - nf^2 = -M_2.$$

**Step 2: Simplifying the Nuclear Norm.** The second term of original objective function can be simplified based on extra equality constraints. We introduce coefficients $A = C_{A1} + nC_{A2}$, $D = C_{D1} + nC_{D2}$, and $B = C_{B1} + nC_{B2}$. From Step 1, we have established that $A = D = M_2$ and $B = -M_2$.

Consequently, the discriminant term $AD - B^2$ becomes:

$$AD - B^2 = (M_2)(M_2) - (-M_2)^2 = M_2^2 - M_2^2 = 0.$$

When the discriminant is zero, the original nuclear norm expression (likely involving square roots of eigenvalues) simplifies significantly, yielding the objective function stated in (27). The remaining constraints are carried over directly, which completes the proof. $\square$

We restate the optimization problem after reformulate and label each constraint to prepare for the optimal solution later.

$$\min_{a_i, b_i, c_i, d_i, e, f, g, h} \quad (n-1)\sqrt{a_1^2 + b_1^2 + c_1^2 + d_1^2 + 2|a_1d_1 - b_1c_1|} \tag{28}$$
$$+ \sqrt{2}\sqrt{(a_1 + na_2)^2 + (b_1 + nb_2)^2 + ne^2 + nf^2},$$

The inequality constraints are

$$g_1 : a_1 - 1 \geq 0,$$
$$g_2 : a_1 + a_2 + 2e - c_1 - c_2 - 1 \geq 0,$$
$$g_3 : b_1 - 1 \geq 0,$$
$$g_4 : b_1 + b_2 - d_1 - d_2 - 1 \geq 0,$$
$$g_5 : d_1 - 1 \geq 0,$$
$$g_6 : d_1 + d_2 - b_1 - b_2 - 2f - 1 \geq 0.$$

The equality constraints are
$$h_1 : a_1 + c_1 + n(a_2 + c_2) = 0,$$
$$h_2 : b_1 + d_1 + n(b_2 + d_2) = 0.$$

To use KKT condition to characterize the solution, we can check that the optimization problem (28) is a convex problem and satisfies the Slater condition which is similar to the proof in Proposition 2. Specific details are omitted here.

Before formally proving Theorem 1, we will prove several propositions that will be helpful in the proof.

**Proposition 3.** *Every optimal solution of reformulated optimization problem (28) satisfy:*

$$a_1 + na_2 = e, \quad c_1 + nc_2 = -e \tag{29}$$

*in which $e \geq 0$.*

*Proof.* Using stationarity property for parameter $c_2, a_2$ and $e$ and the fact $\boldsymbol{u} \neq 0$, we find that the optimal solution satisfies

$$c_2: \quad 0 + \lambda_2 - \mu_1 n = 0 \quad \Rightarrow \quad \lambda_2 = \mu_1 n. \tag{S1}$$

$$a_2: \quad \sqrt{2}\frac{n(a_1 + na_2)}{\sqrt{M_2}} - \lambda_2 - \mu_1 n = 0. \tag{S2}$$

$$e: \quad \sqrt{2}\frac{ne}{\sqrt{M_2}} - 2\lambda_2 = 0. \tag{S3}$$

Substitute Eq. (S1) into Eqs. (S2) and (S3) respectively, we get

$$a_1 + na_2 = e.$$

Another equality comes from the equality constraint $a_1 + c_1 + n(a_2 + c_2) = 0$. Moreover, $e \geq 0$ from Eq. (S3) and $\lambda_2 \geq 0$. □

**Proposition 4.** *For every optimal solution of reformulated optimization problem (28), at least one of the following two inequality constraints is tight:*

$$\begin{aligned} g_1 &: a_1 - 1 \geq 0, \\ g_2 &: a_1 + a_2 + 2e - c_1 - c_2 - 1 \geq 0. \end{aligned} \tag{30}$$

*Proof.* We prove by contradiction. We show that if both two constraints are not tight, there is a feasible perturbation of this optimal point such that object value is strictly smaller. We take

$$\Delta a_1 = -\alpha\varepsilon, \quad \Delta b_1, \Delta b_2, \Delta d_1, \Delta d_2 = 0, \quad \Delta e, \Delta f = 0 \tag{31}$$

By proposition 3, we take the following variables as

$$\Delta a_2 = \frac{\alpha}{n}\varepsilon, \quad \Delta c_1 = -\gamma\varepsilon, \quad \Delta c_2 = \frac{\gamma}{n}\varepsilon. \tag{32}$$

Next we show that by choosing appropriate parameters $\alpha$ and $\gamma$, we can construct a feasible descent direction.

**Step 1: Descent direction.**

To prove that this direction is actually a descent direction, we consider the first-order approximation of the object function. First, We find that $\Delta M_2 = 0$ by our choice. Then we consider case by case:

1. $a_1 d_1 - b_1 c_1 = 0$. In this case, we let $-\alpha\varepsilon d_1 - b_1(-\gamma\varepsilon) = 0$ which implies $\alpha d_1 = b_1\gamma$. As a result, we have

$$\begin{aligned} \Delta M_1 &= 2a_1\Delta a_1 + 2c_1\Delta c_1 \\ &= -2\alpha\varepsilon(a_1 + \frac{c_1 d_1}{b_1}) \end{aligned} \tag{33}$$

   It is a descent direction since $c_1 = \frac{a_1 d_1}{b_1} > 0$ and we can take $\alpha > 0$.

2. $a_1 d_1 - b_1 c_1 > 0$. In this case $M_1 = a_1^2 + b_1^2 + c_1^2 + d_1^2 + 2(a_1 d_1 - b_1 c_1)$. We take $\gamma = 0$ and $\alpha$ sufficiently small and get $\Delta M_1 = -2(a_1 + d_1)\alpha\varepsilon$ which implies that it is a descent direction.

3. $a_1 d_1 - b_1 c_1 < 0$. In this case, we have $M_1 = a_1^2 + b_1^2 + c_1^2 + d_1^2 + 2(b_1 c_1 - a_1 d_1)$ and $c_1 > 0$. We take $\gamma$ sufficiently small and $\alpha = 0$. It is a descent direction.

**Step 2: Feasible direction.**

For inequality constraints $g_1$ and $g_2$, we can take $\varepsilon$ small enough to ensure that they can't hit the boundary. Inequality constraints $g_3$ to $g_6$ and equality constraint $h_2$ still hold since the relevant variables have not changed. Equality constraint $h_1$ still hold due to our choice for $a_1, a_2, c_1, c_2$. As a result, the construction is a feasible direction.

□

Then, we claim that both two inequality constraints must be tight by further analysis. Before we prove this result, we first prove the following proposition.

**Proposition 5.** *There exists $N > 0$ such that, for all $n > N$, every optimal solution of reformulated optimization problem (28) satisfy $c_1^* > 0$.*

*Proof.* For the case of $a_1 d_1 - b_1 c_1 \leq 0$, the result is trivial since $c_1 \geq \frac{a_1 d_1}{b_1} > 0$. And for the case of $a_1 d_1 - b_1 c_1 > 0$, we discuss case by case using Proposition 4.

1. $g_1$ is tight, but $g_2$ is not. Based on complementary slackness of KKT condition, we know that $\lambda_2 = 0$. However, based on Eq. (S1), we get

$$\mu_1 = \frac{\lambda_2}{n} = 0.$$

Then, using the KKT condition with respect to $c_1$:

$$(n-1)\frac{c_1 - b_1}{\sqrt{M_1}} + \lambda_2 - \mu_1 = 0. \tag{34}$$

It implies that $c_1 = b_1 > 0$.

2. $g_2$ is tight, but $g_1$ is not. First, we consider the perturbation of the optimal solution. To maintain the constrains, we let

$$\Delta a_1 = \varepsilon, \Delta a_2 = -\frac{\varepsilon}{n}, \Delta c_1 = \varepsilon, \Delta c_2 = -\frac{\varepsilon}{n}. \tag{35}$$

It is a feasible direction and we consider

$$\Delta M_1 = 2a_1\varepsilon + 2c_1\varepsilon + 2d_1\varepsilon - 2b_1\varepsilon. \tag{36}$$

It implies that

$$a_1 + c_1 + d_1 - b_1 = 0. \tag{37}$$

Otherwise, it will be a descent direction. Using Proposition 3, we get

$$a_2 = \frac{e - a_1}{n}, c_2 = \frac{-e - c_1}{n}.$$

Combine this with the constraint $g_2$, we get

$$\frac{n-1}{n}a_1 + \frac{2n+2}{n}e - \frac{n-1}{n}c_1 = 1. \tag{38}$$

Then, we have

$$1 \leq a_1 \leq \frac{n}{n-1}, 0 \leq e \leq \frac{1}{2n+2}, -\frac{1}{n-1} \leq c_1 \leq 0. \tag{39}$$

Using Eq. (37), we have

$$b_1 - d_1 = a_1 + c_1 \geq 1 - \frac{1}{n-1} \tag{40}$$

We consider

$$M_1 = a_1^2 + b_1^2 + c_1^2 + d_1^2 + 2(a_1 d_1 - b_1 c_1)$$
$$\geq 1 + (2 - \frac{1}{n-1})^2 + 1 + 2 = 8 - O(\frac{1}{n}). \tag{41}$$

Thus the leading term of objective function in this case is at least $2n\sqrt{2}$. Therefore, if a specific construction can be given whose objective function is strictly smaller than the above cases, the proof is complete. The specific construction is given below. We let

$$a_1 = 1, b_1 = 2, c_1 = 0.5, d_1 = 1, e = 0.5, f = -1. \tag{42}$$

And parameters are yet to be determined. Due to our choice, we check constraints $g_2, g_4, g_6$ and $h_1, h_2$ to determine them.

$$g_2 : a_2 - c_2 + 0.5 \geq 0$$
$$g_4 : b_2 - d_2 \geq 0$$
$$g_6 : d_2 - b_2 \geq 0 \tag{43}$$
$$h_1 : 1.5 + n(a_2 + c_2) = 0$$
$$h_2 : 3 + n(b_2 + d_2) = 0.$$

Then we take

$$a_2 = c_2 = -\frac{3}{4n}, b_2 = d_2 = -\frac{3}{2n} \tag{44}$$

Then we compute the objective function

$$M_1 = 6.25, \quad M_2 = 0.75 + 1.25n \tag{45}$$

As a result, the objective function

$$(n-1)\sqrt{6.25} + \sqrt{2}\sqrt{3.75 + 1.25n} < 2\sqrt{2}n. \tag{46}$$

Thus, for large $n$, it makes a contradiction.

3. $g_1$ is tight and $g_2$ is tight. We consider the following perturbation:

$$\Delta a_1 = 0, \Delta c_1 = \varepsilon, \Delta a_2 = 0, \Delta c_2 = -\frac{\varepsilon}{n}, \Delta e = \frac{\varepsilon - \frac{1}{n}\varepsilon}{2} \tag{47}$$

Then we will find that

$$\Delta g_1 = 0, \Delta g_2 = 2\Delta e - \Delta c_1 - \Delta c_2 = 0, \Delta h_1 = 0. \tag{48}$$

Then we consider the change of the objective function:

$$\Delta((n-1)\sqrt{M_1}) = (n-1)\frac{c_1 - b_1}{\sqrt{M_1}}\varepsilon \le -(n-1)\frac{1}{\sqrt{M_1}}\varepsilon \le -\frac{n-1}{2\sqrt{2}}\varepsilon \tag{49}$$

Here we use $\sqrt{M_1} \le 2\sqrt{2}$. If not, we can construct the solution with smaller leading term like the proof in case 2 which implies that the solution is not optimal. Then, we consider the second term:

$$\Delta(\sqrt{2}\|u\|_2) = \sqrt{2}\frac{ne}{\|u\|}\Delta e \tag{50}$$

Using inequality constraint $g_4$ and $g_6$, we get $f \le -1$ which implies $\|u\| \ge \sqrt{n}$. Moreover, we get $e \le \frac{1}{2(n+1)}$ because of $c_1 \le 0$ and $g_1, g_2$ are tight. As a result, we have

$$\Delta(\sqrt{2}\|u\|) \le \frac{\sqrt{2}}{4}\frac{n-1}{(n+1)\sqrt{n}}\varepsilon \tag{51}$$

It implies that

$$\Delta((n-1)\sqrt{M_1} + \sqrt{2}\|u\|) \le -\frac{n-1}{2\sqrt{2}}\varepsilon + O(n^{-\frac{1}{2}})\varepsilon < 0. \tag{52}$$

Thus, we have finished the proof.

$\square$

**Corollary 1.** *For all optimal solution of reformulated optimization problem, the inequality constraint $g_2$ must be tight.*

*Proof.* According to Proposition 4, we consider the case $g_1$ is tight but $g_2$ is not. Based on complementary slackness of KKT condition, we know that $\lambda_2 = 0$. However, based on Eq. (S1), we get

$$\mu_1 = \frac{\lambda_2}{n} = 0.$$

Furthermore, due to Eq. (S2), we get

$$e = \frac{n\sqrt{2}}{\sqrt{B}}(\lambda_2 + \mu_1 n) = 0$$

However, it makes an contradiction since the following constraints are violated due to Proposition 5:

$$a_1 + a_2 + 2e - c_1 - c_2 - 1 = 1 - \frac{1}{n} - c_1 + \frac{1}{n}c_1 - 1 < 0. \tag{53}$$

$\square$

we now give the proof of Theorem 1.

*Proof of Theorem 1.* Thanks to Proposition 5, to prove $q(X, y) > 0$ for all OOD query $(X, y) = ((a_i, r_1, r_2), c_i)$, we just need to show that

$$c_1 + c_2 + g + h > a_1 + a_2 + e + f. \tag{54}$$

This is because

$$s_{(X,y),b_j} = c_1 + c_2 + g + h - \max\{a_1, 0\} - a_2 - e - f, \quad \forall j \in [N]$$
$$s_{(X,y),c_j} = c_1, \quad \forall j \neq i. \tag{55}$$

Here, $c_1 > 0$ by Proposition 5. So we only need to prove $s_{(X,y),b_j} > 0$. Using the constraints $e + g = 0$ and $f + h = 0$, inequality (54) can be reformulated as

$$c_1 + c_2 - (a_1 + a_2) > 2e + 2f. \tag{56}$$

Utilizing Corollary 1, The left side of the inequality is simplified to

$$c_1 + c_2 - (a_1 + a_2) = a_1 + a_2 + 2e - 1 - (a_1 + a_2)$$
$$= 2e - 1. \tag{57}$$

As a result, inequality (56) holds if and only if

$$f < -\frac{1}{2}. \tag{58}$$

However, we get a better upper bound by combining inequality constraints $g_4$ with $g_6$

$$b_1 + b_2 \geq d_1 + d_2 + 1$$
$$\geq b_1 + b_2 + 2f + 2, \tag{59}$$

which implies that

$$f \leq -1. \tag{60}$$

As a result, inequality (56) holds which implies $q(X, y) > 0$ for all OOD query. $\square$

### B.4 PROOF FOR THEOREM 2

We begin our proof with the following lemma, which makes fuller use of the symmetry of the problem.

**Lemma 4** (Existence of symmetry solution). *Suppose $W$ is the solution to the optimization problem 2 without identical task. There exists a solution with $a_1, a_2, b_1, b_2$ and $\alpha, \beta$ such that*

$$W^\intercal = \begin{pmatrix} a_1 I_n + a_2 E_n & b_1 I_n + b_2 E_n & \alpha 1_n & \beta 1_n \\ b_1 I_n + b_2 E_n & a_1 I_n + a_2 E_n & \beta 1_n & \alpha 1_n \end{pmatrix}. \tag{61}$$

*The parameters follow the following constraints:*

$$a_1 \geq 1,$$
$$a_1 + a_2 + \alpha \geq b_1 + b_2 + \beta + 1. \tag{62}$$

*Proof.* We show that some orthogonal transformation of $W^\intercal$ remains a solution to the optimization problem. Let $\sigma$ be an arbitrary permutation of $1, \ldots, n$, and let $P_\sigma \in \mathbb{R}^{n \times n}$ denote the associated permutation matrix. Now, consider a permutation of the logit matrix.

$$\sigma(W^\intercal) = \begin{pmatrix} P_\sigma & 0 \\ 0 & P_\sigma \end{pmatrix} W^\intercal \operatorname{diag}\{P_\sigma, P_\sigma, 1, 1\}.$$

We find that $\sigma(W^\intercal)$ is still an optimal solution of the optimization problem. To verify this, we consider

$$s_{(a_i, r_1), b_j}(\sigma(W^\intercal)) = s_{(a_{\sigma^{-1}(i)}, r_1), b_{\sigma^{-1}(j)}}(W^\intercal) \geq 1, \quad \forall j \in [n] - \{i\},$$
$$s_{(a_i, r_1), c_j}(\sigma(W^\intercal)) = s_{(a_{\sigma^{-1}(i)}, r_1), c_{\sigma^{-1}(j)}}(W^\intercal) \geq 1, \quad \forall j \in [n],$$
$$s_{(b_i, r_1), b_j}(\sigma(W^\intercal)) = s_{(b_{\sigma^{-1}(i)}, r_1), b_{\sigma^{-1}(j)}}(W^\intercal) \geq 1, \quad \forall j \in [n],$$
$$s_{(b_i, r_1), c_j}(\sigma(W^\intercal)) = s_{(b_{\sigma^{-1}(i)}, r_1), c_{\sigma^{-1}(j)}}(W^\intercal) \geq 1, \quad \forall j \in [n] - \{i\}.$$

Moreover, $\sigma(\boldsymbol{W}^\intercal)$ is another solution since orthogonal transformation does not change the nuclear norm. Consider the average over all possible permutations:

$$\frac{\sum_\sigma \sigma(\boldsymbol{W}^\intercal)}{n!} = \left( \begin{array}{cccc} a_1\boldsymbol{I}_n + a_2\boldsymbol{E}_n & b_1\boldsymbol{I}_n + b_2\boldsymbol{E}_n & e\boldsymbol{1}_n & f\boldsymbol{1}_n \\ c_1\boldsymbol{I}_n + c_2\boldsymbol{E}_n & d_1\boldsymbol{I}_n + d_2\boldsymbol{E}_n & g\boldsymbol{1}_n & h\boldsymbol{1}_n \end{array} \right).$$

We further exploit the symmetry and consider the following transformation

$$\tau(\boldsymbol{W}^\intercal) = \left( \begin{array}{cc} 0 & \boldsymbol{I} \\ \boldsymbol{I} & 0 \end{array} \right) \boldsymbol{W}^\intercal \operatorname{diag} \left\{ \left( \begin{array}{cc} 0 & \boldsymbol{I} \\ \boldsymbol{I} & 0 \end{array} \right), \left( \begin{array}{cc} 0 & 1 \\ 1 & 0 \end{array} \right) \right\}.$$

Similar verification shows that $\tau(\boldsymbol{W}^\intercal)$ is still the solution to the optimization problem. Taking the sum of $\boldsymbol{W}^\intercal$ and $\tau(\boldsymbol{W}^\intercal)$, we finish the proof. $\square$

We have $\boldsymbol{1}^\intercal\boldsymbol{W}^\intercal = 0$ similar to the proof of Proposition 2 and consider the problem

$$F = (n-1)\sqrt{2(a_1^2 + b_1^2) + 2|a_1^2 - b_1^2|} + 2\sqrt{(a_1 + na_2)^2 + n\alpha^2} \tag{63}$$

subject to

$$\begin{cases} a_1 \geq 1, \\ a_1 + a_2 + \alpha \geq b_1 + b_2 - \alpha + 1, \\ a_1 + b_1 + n(a_2 + b_2) = 0. \end{cases} \tag{64}$$

To prove Theorem 2, we just need to prove the following condition holds for the optimal solution of optimization problem (63):

$$b_1 + b_2 < a_1 + a_2 \tag{65}$$

*Proof of Theorem 2.* **Step 0: Structural simplification.** Using the identity

$$a^2 + b^2 + |a^2 - b^2| = 2\max\{a^2, b^2\},$$

the first square root in Eq. (63) reduces to

$$\sqrt{2(a_1^2 + b_1^2) + 2|a_1^2 - b_1^2|} = 2\max\{|a_1|, |b_1|\}.$$

Since $a_1 \geq 1 > 0$, the objective becomes

$$F = 2(n-1)\max\{a_1, |b_1|\} + 2\sqrt{X^2 + n\alpha^2}, \qquad X = a_1 + na_2.$$

Meanwhile, the linear constraint rewrites as

$$X = -(b_1 + nb_2).$$

**Step 1. Necessary condition at optimum.** Fixing $(a_1, a_2)$ (so $X$ is fixed), we can vary $(b_1, b_2)$ while preserving $b_1 + nb_2$; this leaves $X$ and the second term unchanged. If $|b_1| > a_1$, then decreasing $|b_1|$ towards $a_1$ reduces the first term and relaxes or maintains the inequality constraint, hence cannot be optimal. Therefore,

$$|b_1| \leq a_1,$$

and the first term simplifies to $2(n-1)a_1$.

**Step 2. KKT analysis in the strict case** $|b_1| < a_1$**.** In this regime, the objective is independent of $b_1, b_2$. We write down the stationary condition for $b_1, b_2$:

$$\begin{aligned} \lambda_2 - \mu_1 &= 0 \\ \lambda_2 - n\mu_1 &= 0. \end{aligned} \tag{66}$$

Thus, we get $\lambda_1 = \mu_1 = 0$. Then we consider stationary condition for $a_2$:

$$2n\frac{X}{\sqrt{X^2 + n\alpha^2}} - \lambda_2 - n\mu_1 = 0. \tag{67}$$

It implies $X = 0$. Finally, we consider $a_1$:

$$2(n-1) + 2\frac{X}{\sqrt{X^2 + n\alpha^2}} - \lambda_1 - \lambda_2 - \mu_1 = 0. \tag{68}$$

As a result $\lambda_1 = 2(n-1)$. Based on complementary slackness, we have $a_1 = 1$. Thus,

$$a_2 = -\frac{1}{n}, \qquad b_2 = -\frac{b_1}{n}, \qquad b_1 + nb_2 = 0.$$

Consequently,

$$a_1 + a_2 = 1 - \frac{1}{n}, \qquad b_1 + b_2 = (1 - \tfrac{1}{n})b_1,$$

and since $|b_1| < 1$, we obtain

$$b_1 + b_2 < 1 - \frac{1}{n} = a_1 + a_2.$$

**Step 3. Boundary case $|b_1| = a_1$.** We first find that $a_1 = 1$. Otherwise, we can perturb $a_1, a_2, b_1, b_2$ to lower the objective function while keeping $X$ fixed. The first term of the optimization object is $2(n-1)$. Moreover, we can consider another solution

$$a_1 = 1, a_2 = -\frac{1}{n}, b_1 = -\frac{1}{n-1}, b_2 = \frac{1}{n(n-1)}, \alpha = 0. \tag{69}$$

Direct verification shows that the constraints are satisfied and the value of the objective function is $2(n-1)$. Thus, if we can find optimal solution in this case, the second term of objective function must be zero. As a result, we have

$$a_1 + na_2 = 0, \alpha = 0. \tag{70}$$

Moreover, it implies that

$$a_2 = -\frac{1}{n}, b_1 + nb_2 = 0 \tag{71}$$

Then we consider the following cases:

(i) $b_1 = -a_1 = -1$. We have $b_2 = \frac{1}{n}$ and $a_1 + a_2 = 1 - \frac{1}{n} > b_1 + b_2$.

(ii) $b_1 = a_1 = 1$. It contradicts with the constraint $a_1 + a_2 + \alpha \geq b_1 + b_2 - \alpha + 1$.

$\square$

## C EXPERIMENTAL DETAILS

### C.1 ARCHITECTURE DETAILS

**Transformer (GPT-2 style).** We use a standard decoder-only transformer with pre-norm residual blocks. Let $d_{\text{vocab}}$ be the vocabulary size, $d_m$ the model width, $d_k$ the per-head query/key dimension, $H$ the number of heads, $L$ the number of layers, and $T$ the context length. Tokens $x_{1:T}$ are embedded by a lookup matrix $\boldsymbol{E} \in \mathbb{R}^{d_{\text{vocab}} \times d_m}$ and summed with learned positional embeddings. Each layer $\ell = 1, \ldots, L$ applies

$$\boldsymbol{Z}^{(\ell)} = \boldsymbol{X}^{(\ell)} + \text{MHA}\Big(\text{LN}\Big(\boldsymbol{X}^{(\ell)}\Big)\Big),$$

$$\boldsymbol{X}^{(\ell+1)} = \boldsymbol{Z}^{(\ell)} + \text{MLP}\Big(\text{LN}\Big(\boldsymbol{Z}^{(\ell)}\Big)\Big),$$

where MHA is causal multi-head attention with $H$ heads (queries/keys/values computed by linear maps in $\mathbb{R}^{d_m \times H d_k}$ and output projection in $\mathbb{R}^{H d_k \times d_m}$), and MLP is a two-layer feed-forward network with GELU activation and hidden size $4d_m$. LayerNorm (LN) is applied in the pre-norm configuration; dropout is disabled unless stated. The language-model head shares weights with $\boldsymbol{E}$ (tied embeddings) and projects to logits in $\mathbb{R}^{d_{\text{vocab}}}$.

### C.2 SYNTHETIC DATASET SETUP DETAILS

**Dataset components.** The components of complexity $= C$ dataset:

(i) (Train) First hop $a \to b$: Input $[a, r_1^c]$, output $[0, b]$, for all $c = 1, \ldots, C$.

(ii) (Train) Second hop $b \to c$: Input $[b, r_2]$, output $[0, c]$.

(iii) (Train) Identity Bridge: Input $[b, ]$, output $[b]$.

(iv) (Test) OOD 2-hop $a \to c$: Input $[a, r_1, r_2]$, output $[0, 0, c]$.

all the sequences are padded to same length by padding token 0.

### C.3 IDENTITY BRIDGE TEST

In this section, we perform a repeat task to monitor for identity bridges that may appear in the language model. Conceptually, the core requirement for the identity bridge is the model's ability to map the input and output of the bridge token $b$. Thus, we use the prompt "Repeat the following token $\{b\}$ directly" with 1-shot example to analyze whether the model possesses an identity bridge-like capability during pre-training. Please note that since the remaining prompts are not specific, completing this task mainly depends on the probability that the token points to itself, so this can be seen as a way to verify the identity mapping. We utilize the Olmo2 (OLMo et al., 2024) 0425 version checkpoints.

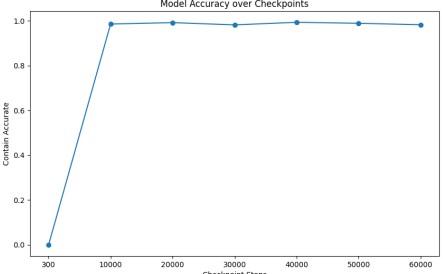

Figure 10: The repeat accuracy of Olmo checkpoints. The repeat entity are set as the bridge token $b$ of MuSiQue dataset.

### C.4 MORE REAL WORLD TASKS

As a supplement, we conducted similar tests on the MuSiQue dataset, which showed a similar mechanism.

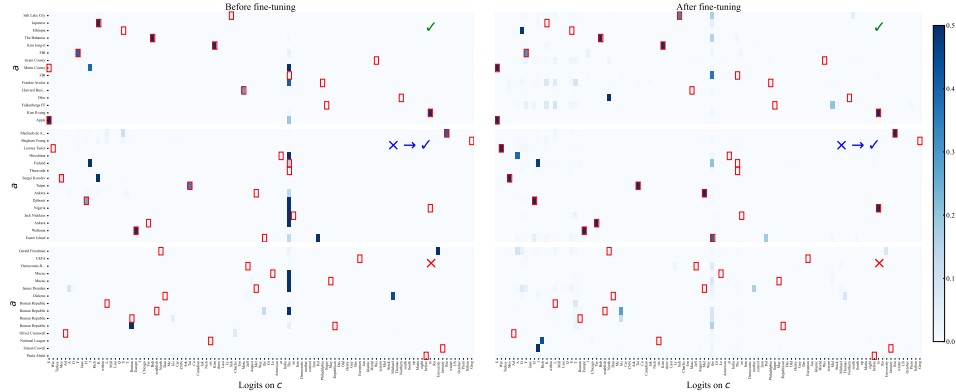

Figure 11: The probability of the model outputting the alternative city token when using the prompt corresponding to $(e_1, r_2)$ before and after fine-tuning on datasets. The red box indicates the answer to the corresponding two-hop reasoning data. The bar chart shows the change in the probability of the corresponding two-hop answer when using the prompt $(e_1, r_2)$ for different datasets.

## D EXPERIMENTS COMPUTE RESOURCES

The experiments were conducted on a server with the following configuration:

- 48 AMD EPYC 7352 24-Core Processors, each with 512KB of cache
- 251GB of total system memory
- 8 NVIDIA GeForce RTX 4080 GPUs with 16GB of video memory each
- The experiments were run using Ubuntu 22.04 LTS operating system

