# OpenReview forum: "Identity Bridge: Enabling Implicit Reasoning via Shared Latent Memory"
_ICLR.cc/2026/Conference — Submitted to ICLR 2026_

### Official Review · Reviewer_rRJu · 2025-10-30

**Soundness:** 2
**Presentation:** 2
**Contribution:** 3
**Rating:** 4
**Confidence:** 3

**Summary:**

This paper explores why LLMs cannot solve what is known as two-hop reasoning ($A → B, B → C$ implies $A → C$). They propose to add during training an identity mapping between intermediate "bridge" entities. The authors argue it improves the model's ability to compose two relations together. They demonstrate this effect on a simple synthetic setup and on real world tasks, demonstrating this phenomenon at model scales up to 8B.

**Strengths:**

The proposed solution is simple and demonstrates significant improvement on two-hop reasoning tasks. The authors relate several interesting ideas, such as connecting grokking to the identity bridge.

**Weaknesses:**

* The notation and presentation of the synthetic setup are difficult to follow. For example, the authors introduce multiple entity and relation sets $\varepsilon_1, \varepsilon_2, \varepsilon_3$ to denote the subject, bridge, and object sets, respectively, and there is to distinguish between the three. The index notation $i$, $j$ is heavily overloaded (it is used to index the entity set $\varepsilon_i$, the elements within the bridge entities $\varepsilon_{2,i}$, the elements within the relation set $r_i$, and so on).

* In Figure 2, the results primarily serve to show that the choice of weight initialization matters for generalization on harder complexity tasks. But I find this setup to be not well motivated and could benefit from further clarification. This finding seems to suggest that two-hop capabiiites can only be unlocked given certain initialization settings. I can see that section 4.4 further explains this, but then I am confused as to why section 4.3 on the Emb-MLP is necessary. In short, I am confused as to what is the natural question that motivates the study of different initializations in Figure 2.

**Questions:**

Two-hop reasoning still feels relatively contrived. How well does this analysis/experiment translate to multi-hop reasoning, beyond just two steps?

---

> ### Author Response · Authors · 2025-11-20
> **Response to Reviewer rRJu**
>
> ### Response to W1:
> We thank the reviewer for this valuable feedback. To improve readability, we have significantly rewritten the Introduction in the revised version to better contextualize the background, related work, and motivation. Furthermore, in the Preliminary section we moved complex notations and formal definitions to Appendix A. This adjustment allows for a much more concise and accessible presentation of the task and model setups in the main text.
> ### Response to W2:
>
> We apologize if the motivation behind Figure 2 and its connection to Section 4.3/4.4 was unclear. We have revised the manuscript to better articulate the logical flow. Here, we clarify the relationship between 2-hop reasoning, Emb-MLP, and initialization through the lens of two distinct mechanisms:
>
> 1. Theoretical Context: Theoretically, performing strict step-by-step inference for an N-hop task typically requires a network depth of N−1. This suggests a single-layer Transformer should fail at 2-hop reasoning. However, previous observations of OOD failures imply that Transformers do not always rely on strict step-by-step logic.
>
> 2. Mechanism 1: The Shortcut (Section 4.3): We found that adding an identity bridge allows even a single-layer Transformer to solve 2-hop tasks. We explain this via the Emb-MLP theory: the bridge helps the model form a direct "shortcut" (a→c) without intermediate reasoning steps. This mechanism is prevalent in pre-trained LLMs.
>
> 3. Mechanism 2: The Alignment (Section 4.4): However, relying solely on the shortcut (identity bridge + depth) is insufficient for robust OOD generalization (accuracy often remains capped at 1/complexity in Figure 2). While deep models have the capacity for true step-by-step reasoning (Alignment), they do not automatically adopt it. This is the natural question driving Figure 2: How do we unlock this deeper capacity?
>
> Conclusion: Figure 2 demonstrates that specific regularization strategies,  such as small initialization [1,2] and weight decay (added in the revision), are necessary to force the model to align hidden states and learn the robust "Alignment" mechanism, rather than settling for the easier "Shortcut." Thus, we identify two regimes: the Shortcut (explained by Emb-MLP) and the Alignment (unlocked by regularization/initialization).
>
> ### Response to Q1:
>
> We appreciate the reviewer’s suggestion to extend our analysis beyond the 2-hop setting. In response, we have added new experimental results and detailed discussions regarding 3-hop reasoning in Section 4.5 of the revised manuscript.
>
> Our observations indicate that the behavioral patterns in 3-hop tasks closely mirror those found in the 2-hop scenarios. This consistency suggests that the mechanisms we identified (such as the interplay between shortcuts and alignment) are not artifacts of the 2-hop setup, but rather scalable properties that generalize to longer reasoning chains.
> ### Reference
>
> [1] Yao, Junjie, Zhongwang Zhang, and Zhi-Qin John Xu. "An Analysis for Reasoning Bias of Language Models with Small Initialization." Forty-second International Conference on Machine Learning.
>
> [2] Zhongwang Zhang, Pengxiao Lin, Zhiwei Wang, Yaoyu Zhang, Zhi-Qin John Xu*, Initialization is Critical to Whether Transformers Fit Composite Functions by Reasoning or Memorizing, NeurIPS 2024

---

> ### Author Response · Authors · 2025-11-27
>
> Dear Reviewer rRJu,
>
> We hope this message finds you well. Thank you again for your insightful review of our paper. Regarding the concerns you raised in your review, including the notations, the motivation for small initialization, the extension of the experiment, we have provided specific and detailed responses in the rebuttal. We would be very grateful to know whether these responses have addressed your concerns. If you have any further questions about our work or our responses during the discussion period, please do not hesitate to let us know.
>
> Once again, thank you for your valuable time and feedback.
>
> Best regards,
> The Authors

---

### Official Review · Reviewer_dojw · 2025-10-31

**Soundness:** 3
**Presentation:** 2
**Contribution:** 2
**Rating:** 4
**Confidence:** 3

**Summary:**

The paper addresses a core limitation of LLMs, failure at compositional reasoning, particularly two-hop reasoning.
It introduces Identity Bridge, a simple zero-hop auxiliary task that enforces latent-space alignment between bridge entities and improves out-of-distribution (OOD) two-hop generalization.

**Strengths:**

- Conceptual clarity. The “identity bridge” idea is strikingly simple yet yields strong intuition about latent geometry and shared memory.
- Theoretical-empirical coherence. The nuclear-norm analysis provides a plausible mechanistic link between auxiliary supervision and OOD reasoning.
- Good connection to literature. Relates to grokking, implicit bias, and compositional generalization.
- Readable presentation. Figures 3–8 visualize reasoning patterns and hidden-state alignment.

**Weaknesses:**

1. Novelty scope. The “identity bridge” can be viewed as a trivial self-supervised auxiliary loss; the work should better distinguish it from prior identity-based pretraining (e.g., autoencoding, copy tasks).
2. Theoretical depth. The Emb-MLP uniform-attention analysis is elegant but may be too idealized; discussion should address limitations of extending results to real transformers.
3. Empirical diversity. All tasks are synthetic or limited to the TWOHOPFACT dataset; a real-world compositional benchmark would greatly strengthen claims.
4. Interpretation of LLM results. The “implicit identity bridge during pretraining” claim (Sec. 4.5) is speculative; more ablations or probing experiments are needed to support this inference.

**Questions:**

See above

---

> ### Author Response · Authors · 2025-11-20
>
> ### Response to W1: Novelty scope of the identity bridge
>
> We respectfully wish to clarify the scope and primary contribution of our work, as there appears to be a misunderstanding regarding the role of the "identity bridge."
>
> Our core contribution is not to propose the identity bridge as a novel pre-training objective or method. Instead, our goal is to explain why transformer often fails at implicit reasoning. We reveal that the model's performance is heavily dependent on shortcuts. When task complexity increases, this leads to low accuracy as shown in Figure 2.
>
> In this context, the identity bridge serves as an analytical instrument to verify our mechanistic understanding on synthetic data. While we acknowledge that the loss formulation mathematically resembles standard autoencoding, its role here is fundamentally different.
>
> ### Response to W2: Theoretical depth
>
>
> We acknowledge that our theoretical analysis relies on an idealized one-layer setting. We chose this minimal model because it is sufficient to capture the core mechanism of the "shortcut" in two-hop reasoning, allowing for a rigorous mathematical derivation that was lacking in prior work.
>
> While we recognize that real transformer are far more complex, our empirical experiments (Figure 3) demonstrate that the theoretical insights derived from this simplified model extrapolate well to complex architectures. The Emb-MLP theory provides the mechanistic blueprint, which we then verify in practice.
>
> To explicitly address the gap between theory and practice, we have rewritten the Limitations section in the revised manuscript to provide a more nuanced discussion on the constraints of our theoretical assumptions and the extent of their applicability to deep Transformers.
>
> ### Response to W3: Real-world benchmark
> We would like to clarify that TWOHOPFACT is a real-world compositional benchmark: it is constructed from factual triples (e.g., novel–author–birthcity) derived from real-world knowledge bases, and the two-hop queries involve genuine entities and relations rather than purely synthetic symbols. Thus, our evaluation is not restricted to fully synthetic toy tasks.
>
> ### Response to W4: Interpretation of LLM results
> We appreciate the reviewer’s concern and agree that our current wording may sound stronger than what our experiments strictly justify. Our intention in Sec. 4.5 is to provide a mechanistic interpretation of the observed LLM behavior, not to claim a fully proven statement about pretraining dynamics.
>
> Importantly, in the LLM experiments we only fine-tune on single-hop data. If the pretrained model had no structural constraint or inductive bias on the bridge entities, then after fine-tuning it should remain essentially incapable of two-hop reasoning, similar to a model trained from scratch. However, we observe that:
>
> * Fine-tuning a pretrained model on single-hop data does improve its two-hop performance.
>
> * This improvement relies primarily on the shortcut from $a$ to $c$, mirroring the shortcut mechanism we characterize in the Emb-MLP analysis.
>
> It implies that pretraining has already instantiated an identity-like bridge that our fine-tuning then leverages.

---

> ### Author Response · Authors · 2025-11-27
>
> Dear Reviewer dojw,
>
> We hope this message finds you well! Thank you for your insightful review of our paper. As for the concerns about our work in the review, we have provided very specific and detailed responses. Have these responses resolved your concerns? If you have any further questions about our work or responses during this discussion period, please do not hesitate to contact us. We sincerely look forward to receiving your further comments on our responses.
>
> Once again, thank you for your valuable time and feedback!
>
> Best regards,
>
> Authors

---

### Official Review · Reviewer_ty5k · 2025-10-31

**Soundness:** 2
**Presentation:** 2
**Contribution:** 2
**Rating:** 2
**Confidence:** 3

**Summary:**

The paper studies a phenomenon in LLMs reasoning where models often fail to learn "A to C" from "A to B" and "B to C", which is termed as the curse of two-hop reasoning. One understanding is that the latent representation of B from the first hop does not necessarily aligns with the representation from the second hop. The paper proposes a simple fix, called identity bridge, by adding zero-hop identity "B to B" to bridge this gap. Empirical results on synthetic data seem promising. Theoretical understanding is also provided using a simplified linear model, showing that the identity bridge induces nuclear norm regularization.

**Strengths:**

The paper presents a clear idea that tries to address the curse of two-hop reasoning. The proposed mechanism is simple and interpretable.  An intuitive theoretical link is provided to better understand how this simple fix helps with the two-hop reasoning phenomenon. Empirical results on synthetic data match with theoretical insights.

**Weaknesses:**

1. I am still confused why a new mechanism is required given that CoT already works well. The paper discussed some shortcomings of CoT, but all of them are not verified in this paper's setting. I can imagine that CoT would work perfectly on both the synthetic data and real-world data. Given that the accuracy improvement brought by the identity mapping is not significant on real-world dataset, I would still do CoT compared with the identity bridge. The motivation of this paper to study the identity bridge needs further explanation.

2. The claim on implicit nuclear norm regularization is not mathematically rigorous. Eq. (1) and Eq. (2) are two different optimization problems, and they have different solutions. It is not correct to just study Eq. (2) as a reformulation of Eq. (1). There is no formal and rigorous proof on the equivalence between the two but only some empirical observations on synthetic data. This is not enough to just claim identity bridge induces implicit nuclear norm regularization.

3. The statements of Theorem 1 and 2 are not precise. In particular, important assumptions such as Assumptions 1-4 are not even mentioned and discussed in the statements. This is misleading and gives a sense that the theoretical results are very general. In fact, they only work for very limited cases. For example, why can we just assume $1^\top W=0$ in Assumption 2? This is never verified and particularly important for the theorems to be true. Therefore, I think the theoretical results are a bit overclaiming.

4. More baselines and ablation studies could be added. Are the empirical results stable against randomness, e.g., change a different random seed?

**Questions:**

See weaknesses.

---

> ### Author Response · Authors · 2025-11-20
> **Response to Reviewer ty5k (1/2)**
>
> We thank the reviewer for the careful reading and constructive comments. We address each concern below.
>
> ### Weakness 1. Motivation and relation to CoT
> We appreciate the reviewer’s comment and realize that our original writing may have given a misleading impression. Our goal is not to propose a new method to replace chain of thought (CoT), nor to claim that identity bridge is a better method. In fact, our work focus on understanding the mechanism of implicit reasoning in transformers.
>
> Recent research shows that Large Language Models (LLMs) have shortcomings in implicit reasoning, especially when faced with out-of-distribution tasks [1,2]. However, from the perspective of approximation, overparameterized models are fully capable of implicit inference. This gap between capability and actual OOD behavior in the implicit regime is the main motivation for our study.
>
> Our contribution is to demonstrate that, after appropriately adjusting the training data, such as by adding an identity bridge, even a single-layer Emb-MLP model can learn OOD two-hop generalization. This indicates that transformers can solve multi-hop tasks via a very simple shortcut mechanism rather than what would normally be considered "genuine" implicit reasoning. We empirically characterize this learning mechanism and theoretically demonstrate why the identity bridge works and why its absence leads to generalization failure.
>
> We agree that CoT can work very well on both our synthetic and real-world datasets, and we do not dispute that. However, explicit CoT and implicit reasoning are complementary: CoT is an external prompting strategy, while we consider the internal representation of the model and how to enable implicit reasoning. Furthermore, implicit reasoning also has its potential advantages. Mechanistically, this means the model learns better internal representations. In application, it can significantly reduce the number of generated tokens and improve inference speed. Moreover, it avoids the additional step of parsing the final answer from a long chain of intermediate tokens, which can sometimes harm instruction following and robustness.
>
> We will clarify these points in the introduction and motivation sections to dispel any misunderstandings about potential competition with CoT.
>
> ### Weakness 2. Equivalence between Eq. (1) and Eq. (2)
> We appreciate the reviewer’s concern and agree that our original presentation did not make the mathematical connection sufficiently precise. But in fact, these two optimization problems are completely equivalent. Regarding the equivalence of these two problems, please refer to Lemma 1 in [3]. The idea is that reparameterizing $E$ and $W_\mathrm{proj}$ as $W$ will result in the following relationship between the nuclear norm and the Frobenius norm.
>
> $$
> \Vert W \Vert_*^2 = \min\_{\{ E W\_{\mathrm{proj}} = W\}} \frac{1}{2} \left( \Vert E\Vert \_{\mathrm{F}}^2 + \Vert W\_{\mathrm{proj}} \Vert \_{\mathrm{F}}^2\right)
> $$
>
> Meanwhile, in the Emb-MLP model, we have
> $$
> \frac{1}{2} \Vert \theta \Vert\_2^2  = \frac{1}{2} \left( \Vert E\Vert\_{\mathrm{F}}^2 + \Vert W\_{\mathrm{proj}}\Vert\_{\mathrm{F}}^2\right)
> $$
> These two relations demonstrate that the two optimization problems are equivalent. That is, for the solution $(E, W_\mathrm{proj})$ of optimization problem (1), their reparameterization is the solution of optimization problem (2). Conversely, for the solution $W$ of optimization problem (2), there is a way to split it into its corresponding $(E, W_\mathrm{proj})$.
>
> In the revised edition, we describe the equivalence and ideas of these two problems and provide appropriate references.

---

> ### Author Response · Authors · 2025-11-20
> **Response to Reviewer ty5k (2/2)**
>
> ### Weakness 3. Precision of Theorem 1 and 2 and role of assumptions
> We agree that the theorem statements in the current version are not sufficiently precise and may rely too heavily on assumptions.
>
> In the revised version, we reconsidered the entire proof process and introduced the language of subdifferentials. Through these efforts, we greatly reduced our reliance on the proof, retaining only Assumption 1, which was introduced in reference [3]. Next, we will comment on each hypothesis and its modification.
>
> 1. About Assumption 1. Assumption 1 is a symmetry condition introduced in reference [3], which assumes that the parameters of the network also have symmetry, taking into account the symmetry of the data. Therefore, for solutions that violate symmetry, the condition $\mathbb{1}^{\intercal} W^\intercal \neq 0$ is introduced. It can be seen that this inequality condition is relatively loose. Intuitively, due to the symmetry of the data, it is almost impossible for the network to learn an asymmetric solution and strictly satisfy $\mathbb{1}^{\intercal} W^\intercal = 0$. Therefore, this assumption is reasonable.
>
> 2. About Assumption 2. Assumption 2 has now been rigorously formalized into Proposition 2. Its proof utilizes the symmetry of the problem.
>
> 3. About Assumption 3. Assumption 3 is no longer necessary. We introduced Assumption 3 because we weren't using the language of subdifferentials, which required some relaxation of the problem, causing the original convex problem to become non-convex. Now, consider the original problem, which is convex and satisfies the Slater condition. Therefore, all optimal solutions satisfy the KKT condition.
>
> 4. About Assumption 4. Assumption 4 has now been replaced by Proposition 5 and Corollary 1. In Proposition 5, we proved that the first part of Assumption 4, $c_1 > 0$, is strictly true. The second part of Assumption 4, after our reconsideration, is actually not necessary for the proof of Theorem 1.  Therefore, this assumption can be simplified.
>
> In addition, in the revised version, we have adjusted the statement of the theorem to make it as rigorous as possible.
>
> ### Weakness 4. Baselines, and randomness
> We agree that our empirical presentation can be clearer. However, at the same time, we reiterate that our work is not to prove that the identity bridge is superior to CoT, but rather to elucidate the mechanism of implicit reasoning. Therefore, the experimental focus should be on the impact of the presence or absence of an identity bridge on two-hop reasoning ability.
>
> 1. Baselines. In Figure 2, we already compare the Emb-MLP model with and without the identity bridge. Without the identity bridge, the model consistently fails to learn OOD 2-hop reasoning, mirroring the OOD failure modes reported in [2]. We will make this baseline more prominent in the main text and caption, explicitly referring to “no-bridge” as the baseline model.
>
> 2. Stability across random seeds. All experiments in Figure 2 are run with five different random seeds, and the error bars correspond to the standard deviation across these runs.
>
> ### Summary
> We sincerely hope that our explanations regarding the article's motivation, theoretical rigor, and experimental stability will address your concerns and reverse negative opinions you may have about the article.
>
> ## Reference
> [1] Boshi Wang, Xiang Yue, Yu Su, and Huan Sun. "Grokking of implicit reasoning in transformers: A mechanistic journey to the edge of generalization." Advances in Neural Information Processing Systems, 37, 2024.
>
> [2] Jiaran Ye, et al. "How does Transformer Learn Implicit Reasoning?" Advances in Neural Information Processing Systems, 2025.
>
> [3] Huang, Yixiao, et al. "Generalization or Hallucination? Understanding Out-of-Context Reasoning in Transformers." Advances in Neural Information Processing Systems, 2025.

---

> > ### Author Response · Authors · 2025-11-27
> >
> > Dear Reviewer ty5k,
> >
> > We hope this message finds you well. Thank you again for your insightful review of our paper. Regarding the concerns you raised in your review, including the motivation of the paper, the equivalence between optimization problems, and the dependence on assumptions, we have provided specific and detailed responses in the rebuttal. We would be very grateful to know whether these responses have addressed your concerns. If you have any further questions about our work or our responses during the discussion period, please do not hesitate to let us know.
> >
> > Once again, thank you for your valuable time and feedback.
> >
> > Best regards,
> >
> > The Authors

---

### Official Review · Reviewer_f5MC · 2025-11-05

**Soundness:** 3
**Presentation:** 3
**Contribution:** 3
**Rating:** 4
**Confidence:** 4

**Summary:**

This paper considers the task of 2-hop reasoning: Given an entity e1, and relations r1 and r2, the objective is to find tail entity e_answer by following the reasoning (e1, r, e_bridge) and (e_bridge, r2, e_answer). This task is commonly hard for LLMs without chain-of-thought reasoning, especially when the pair (r1, r2) is not observed during training time. To improve such OOD 2-hop reasoning, the paper introduces “identity bridge” tasks where the model has to simply implement the identity function over bridge entities.

In addition to experimenting with transformers the paper introduces a simpler neural family, “Emb-MLP” which is a simple BoW based predictive model. Initial results show that both transformers and Emb-MLP have similar performance on the 2-hop OOD reasoning task, and thus the paper analyzes Emb-MLP further.

**Strengths:**

- Results show that this additional auxililary training objective dramatically improves OOD generalization on 2-hop tasks. By training on the bridge identity task, geometry of the embeddings for Emb-MLP task shifts in a way that supports OOD generalization.
- Identifies a practical intervention (identity supervision) that addresses a known failure mode and provides both empirical and theoretical characterization of the phenomenon
- The theoretical analysis (Theorems 1-2) for the Emb-MLP model is rigorous, with detailed proofs
- Experimental validation across multiple complexity levels shows consistent patterns

**Weaknesses:**

- The theory relies heavily on Assumptions 1-4, particularly the regularity and non-degeneracy assumptions (3-4), which are not empirically validated
- The gap between the uniform-attention Emb-MLP theory and actual transformer behavior is acknowledged but not bridged
- The real-world LLM experiments (Section 4.5) are more correlational than causal—they show probability shifts consistent with the theory but don't definitively prove the mechanism

**Questions:**

- Do learned solutions empirically exhibit the non-degeneracy properties assumed?
- How sensitive are the results to the specific identity task formulation? What if identity mapping is noisy or only partially supervised?
- The paper claims identity bridges emerge during pretraining—can you provide more direct evidence (e.g., probing experiments on models at different pretraining checkpoints)?

---

> ### Author Response · Authors · 2025-11-20
> **Response to Reviewer f5mc (1/2)**
>
> We thank the reviewer for the careful reading and constructive comments. We will address the weaknesses and problems one by one in the following text.
> ### Response to W1: Role and validation of Assumptions 1–4
>
> In the revised version, we have substantially reworked the theoretical section to both (i) make the statements of Theorems 1–2 more precise and (ii) greatly reduce the reliance on strong assumptions (from 4 assumptions to 1). We will explain each point based on the assumptions and their modifications.
>
> 1. **Assumption 1 (symmetry): Kept**
>    Assumption 1 is a symmetry condition introduced in prior work [1]. Starting from the symmetry of the data itself, it excludes asymmetric solutions with measure zero by requiring $(\mathbf{1}^\intercal W^\intercal \neq 0)$. This inequality is very mild: given the symmetric data distribution, it is highly unlikely that optimization converges to a non-symmetric but strictly degenerate solution with $(\mathbf{1}^\intercal W^\intercal = 0)$. We keep Assumption 1, and we now state explicitly that it is inherited from [1] and only excludes a pathological set of solutions.
>
> 2. **Assumption 2 (structural property): Reduced**
>    What was previously stated as Assumption 2 has now been upgraded to a formal statement as **Proposition 2**, with a complete proof in the appendix. The proof uses the symmetry of the problem and no longer relies on additional regularity assumptions. Thus, this condition is no longer an unverified assumption but a derived property of the objective.
>
> 3. **Assumption 3 (regularity): Reduced**
>    The regularity assumption is **no longer needed**. It was only introduced in the original version because we avoided subdifferential notation and instead worked with a relaxed, slightly non-convex surrogate of the optimization problem. In the revised proof, we work directly with the original convex problem and its subdifferentials, which, together with Slater’s condition, guarantees that all optimal solutions satisfy KKT conditions. This makes the previous Assumption 3 redundant, and we have removed it entirely.
>
> 4. **Assumption 4 (non-degeneracy): Reduced**
>    Assumption 4 has been replaced by **Proposition 5** and **Corollary 1**. In Proposition 5 we prove that the crucial part of the previous assumption, namely $c_1 > 0$, actually holds for every optimal solution and is therefore not an assumption but a proposition. After re-examining the proof of Theorem 1, we realized that the remaining part of Assumption 4 was never essential; accordingly, it has been dropped.
>
> ---
>
> ### Response to W2: Gap between Emb-MLP theory and actual transformer behavior
>
> The reviewer is correct that there is a conceptual gap between Emb-MLP and transformers. Our goal is not to claim that a real multi-layer transformer is literally identical to Emb-MLP, but that Emb-MLP captures the **core mechanism** by which identity bridges are implemented.
> First, the Emb-MLP serves as a valid approximation of a uniform-attention Transformer, a modeling approach widely adopted in prior literature [3-5]. We specifically chose to establish our theory on Emb-MLP after observing that 2-hop reasoning tasks can be resolved without the participation of attention. Second, regarding multi-layer scenarios, our experiments demonstrate that the performance patterns of actual Transformers align closely with those of Emb-MLP across varying levels of complexity (Figure 2). Furthermore, the similarity observed in the row-wise logit templates between EMB-MLP and GPT-2 (Figure 3) corroborates this alignment. These findings confirm that the mechanistic principles identified in Emb-MLP remain operative and preserved within multi-layer Transformers.
>
> ---
>
> ### Response to W3: Correlational nature of the real-world LLM experiments
>
> We appreciate the reviewer's comment and realize that our original writing may have given a misleading impression. Therefore, we clarify our reasoning. We have theoretically established two distinct mechanisms for learning 2-hop reasoning: the shortcut and the alignment. To verify the specific existence of the shortcut mechanism in real-world scenarios, we designed a controlled intervention experiment. Specifically:
>
> 1. We fine-tuned the model using **only** single-hop data $(a, r_1, b)$ and $(b, r_2,c)$.
>
> 2. We evaluated the model's response to the query $(a, r_2, )$ before and after fine-tuning to see if the $a\to c$ shortcut could be established without any identity bridges.
>
> We observed a significant probability shift for the target $c$ after fine-tuning. This result provides evidence for the formation of the $a\to c$ shortcut, confirming its existence beyond mere correlation.

---

> ### Author Response · Authors · 2025-11-20
> **Response to Reviewer f5mc (2/2)**
>
> ---
>
> ### Response to questions
>
> **Q1. Do learned solutions empirically exhibit the non-degeneracy properties assumed?**
>
> In the revised theory, the key non-degeneracy properties are no longer assumptions:
>
> * The crucial condition ($c_1 > 0$) is proven as part of **Proposition 5**.
> * The remaining structural properties previously grouped under Assumption 4 are unnecessary.
>
> Therefore, our main theorems no longer rely on an unverified non-degeneracy assumption about the learned solution.
>
> ---
>
> **Q2. Sensitivity to the identity task formulation (noisy / partially supervised identity mapping).**
>
> We have provided noisy / partially supervised identity mapping experiments in revised Figure 2 (right). The performance of these experiments are meet our expectations.
>
> ---
>
> **Q3. Direct evidence that identity bridges emerge during pretraining.**
> First, we wish to clarify that the discussion regarding the emergence of identity bridges during pretraining was intended as a theoretical hypothesis rather than a formal claim. We acknowledge that direct probing evidence was not the central focus of our original manuscript.
>
> However, to address your query and offer preliminary empirical support, we conducted a simplified analysis using the intermediate checkpoints of OLMo-2-0425-1B [2]. Conceptually, the core requirement for an identity bridge is the model's ability to map the input and output of the bridge token b (linking the generated $b$ from $a→b$ to the input $b$ for $b→c$). A straightforward proxy for this capability is the "repeat task" (i.e., testing if the model can "Directly repeat the {$b$}").
>
> Our tests indicate that the model acquires this repeating capability at a very early stage of pretraining (around Step 10,000). While this is an indirect measure, it suggests that the foundational mechanism required for identity bridges is likely a common and early-emerging property in real-world LLMs.
>
> ### Reference
> [1] Huang, Yixiao, et al. "Generalization or Hallucination? Understanding Out-of-Context Reasoning in Transformers." Advances in Neural Information Processing Systems, 2025.
>
> [2] OLMo, Team, et al. "2 OLMo 2 Furious." arXiv preprint arXiv:2501.00656 (2024).
>
> [3] Yao, Junjie, Zhongwang Zhang, and Zhi-Qin John Xu. "An Analysis for Reasoning Bias of Language Models with Small Initialization." Forty-second International Conference on Machine Learning.
>
> [4] Mahankali, Arvind V., Tatsunori Hashimoto, and Tengyu Ma. "One Step of Gradient Descent is Provably the Optimal In-Context Learner with One Layer of Linear Self-Attention." The Twelfth International Conference on Learning Representations.
>
> [5] Zhang, Ruiqi, Spencer Frei, and Peter L. Bartlett. "Trained transformers learn linear models in-context." Journal of Machine Learning Research 25.49 (2024): 1-55.

---

> ### Comment · Reviewer_f5MC · 2025-11-21
> **Thank you for the detailed response + questions**
>
> Thank you for the detailed response, the new experiments and changes to the assumptions and the proofs.
>
> I have a few additional follow-up questions:
>
> 1. >"we conducted a simplified analysis using the intermediate checkpoints of OLMo-2-0425-1B [2]...A straightforward proxy for this capability is the "repeat task" (i.e., testing if the model can "Directly repeat the {$b$}")"
>
> Could you please add this experiment to the paper and provide a bit more detail. It is not clear to me how the "repeat" task would be any indication of the emergence of an identity bridge during pre-training.
>
> 2. How would you apply the findings in this paper to show improvement on some real world benchmarks that require >= 2-hop reasoning?

---

> > ### Author Response · Authors · 2025-11-22
> > **Response to additional questions**
> >
> > **Q1: Details of repeat task**
> >
> > We have supplemented the relevant experimental settings and results in Appendix C.3. These show that the model demonstrates replicability early in the pre-training process. Please note that since the prompts are not specific, completing this task mainly depends on the probability that the token points to itself, so this can be seen as a way to verify the identity mapping.
> >
> > **Q2: Improvement on benchmark**
> >
> >
> > We thank the reviewer for the question. We clarify that our primary goal is mechanistic explanation rather than performance improvement.
> >
> > 1. Our goal is to explain why vanilla Transformers fail to generalize on OOD two-hop instances and what minimal changes can unlock generalization capabilities. We established that the "identity bridge" facilitates a shortcut mechanism although it is a key component to unlocking two-hop reasoning capabilities. We empirically demonstrated that models become "trapped" by these shortcuts, which prevents them from acquiring the correct, robust step-by-step reasoning strategies.
> >
> > 2. We found that real-world LLMs exhibit this exact failure mode. This is likely the reason why current large models remain inefficient at implicit reasoning.
> >
> > Consequently, our contribution is to reveal the underlying mechanism of composite reasoning. We do not propose a new method to improve benchmark scores in this work. We have verified that this mechanism also applies to 3-hop scenarios in Section 4.5.

---

> > ### Author Response · Authors · 2025-11-27
> >
> > Dear Reviewer f5MC,
> >
> > We hope this message finds you well. Thank you again for your insightful review of our paper. Regarding your additional questions, including the experimental details and the discussion of the benchmark, we have provided specific and detailed responses in our reply. We would be very grateful to know whether these responses have addressed your concerns. If you have any further questions about our work or our responses during the discussion period, please do not hesitate to let us know.
> >
> > Once again, thank you for your valuable time and feedback.
> >
> > Best regards,
> >
> > The Authors

---

### Author Response · Authors · 2025-12-02
**Global response (1/2)**

Dear Area Chair,

we briefly summarize below the motivation and contributions of our work, and how the revised version addresses the main concerns raised in the reviews.

---
**Motivation.**
Recent work suggests that transformers have intrinsic limitations in implicit reasoning. On synthetic datasets, they often completely fail to generalize on out-of-distribution (OOD) reasoning tasks. On real-world tasks, empirical analyses show that implicit reasoning is fragile and dominated by shortcuts.

This indicates that, although transformers are expressive enough in principle to implement implicit reasoning, standard data and gradient-based training methods often don't endow them with this capability. Understanding the mechanisms of implicit reasoning is therefore crucial for understanding how transformers actually reason.

---

**Contributions.**
We introduce the identity bridge, a minimal modification to the training data that enables OOD compositional generalization. To understand the underlying mechanism, we develop a uniform-attention theory (Theorems 1 and 2), proving that adding an identity bridge unlocks OOD generalization, while its absence leads to failure.

Experimentally, we show that the simplified Emb–MLP model closely matches the behavior of a standard GPT-2 transformer (Figures 2 and 3), supporting the applicability of our theory. Moreover, the theory predicts that standard transformers implement implicit reasoning via shortcut solutions, which is consistent with prior work and our own observations on real-world tasks.

---

We have provided detailed, point-by-point responses to all reviewer comments and revised the manuscript accordingly. Here we summarize several key questions and their responses.

### 1. Clarifying the motivation (ty5k, dojw, rRJu)

We substantially rewrote the Introduction. The revised version summarizes empirical phenomena and open questions from the existing literature, formulates our central research question explicitly, and lays out the full research approach. We believe this resolves the concern about unclear motivation.

### 2. Strength of theoretical assumptions (f5MC, ty5k)

In the revision, we revisited the entire proof strategy and rewrote the arguments using subdifferential language. This allowed us to significantly weaken and streamline the technical assumptions: the only external assumption we retain is Assumption 1 from [1]. The other conditions that were previously labeled Assumptions 2–4 are now either derived as propositions or shown to be unnecessary.

- **Assumption 1.**
  Assumption 1 is the symmetry condition introduced in [1]. For solutions that break this symmetry, they introduce the inequality constraint $\mathbf{1}^\top W^\top \neq 0$. This inequality is quite mild. Hence we view Assumption 1 as a reasonable and weak condition.

- **Assumption 2.**
  The content previously stated as Assumption 2 is now formalized as **Proposition 2**, whose proof relies only on the symmetry structure of the problem.

- **Assumption 3.**
  Assumption 3 is no longer needed. It was introduced in the original submission because we did not yet use subdifferential tools.

- **Assumption 4.**
  Assumption 4 has been replaced by **Proposition 5** and **Corollary 1**. In Proposition 5, we rigorously prove the first part of what was previously Assumption 4; this part is mathematically correct as stated. Upon re-examining the proofs, we realized that the second part of Assumption 4 is in fact not needed for the proof of Theorem 1, so the assumption can be simplified accordingly.

We have also tightened the statements of the theorems throughout to make the theory as rigorous and explicit as possible.


### 3. Gap between Emb–MLP theory and real transformers (f5MC, dojw)

The reviewers correctly point out that there is a conceptual gap between Emb–MLP and full multi-layer transformers. Our goal is *not* to claim that a real multi-layer transformer is identical to Emb–MLP, but rather that Emb–MLP captures the *core mechanism* by which the identity bridge operates.

- First, Emb–MLP can be viewed as a good approximation of a **uniform-attention transformer**, a modeling abstraction that has been used in prior work. We chose Emb–MLP because we empirically observed that the two-hop reasoning task can be solved even without an explicit attention mechanism.
- Second, in the multi-layer setting, our experiments show that the performance patterns of real transformers closely track those of Emb–MLP across different levels of task complexity (Figure 2).
- Third, the row-wise logit templates of Emb–MLP and GPT-2 are strikingly similar (Figure 3), further supporting that they share the same underlying mechanism.

Taken together, these results indicate that the mechanism identified in Emb–MLP is preserved and remains operative in standard multi-layer transformers.

---

### Author Response · Authors · 2025-12-02
**Global response (2/2)**

### 4. Additional empirical evidence (f5MC, ty5k, dojw, rRJu)

- **Pretraining evidence for identity bridge (f5MC).**
  Reviewer f5MC asked for more direct evidence that identity-bridge-like structure arises during pretraining. In response, we added experiments using **OLMo-2 checkpoints** and a repeat task, showing that the model already forms an effective identity bridge early in pretraining.

- **Baselines and repetitions (ty5k).**
  Reviewer ty5k requested more baselines and repeated runs. We emphasize that our main comparison is *not* against various chain-of-thought methods, but between models **with** and **without** an identity bridge. As shown in Figure 2, removing the identity bridge leads to complete OOD failure. Moreover, Figure 2(a) now reports the mean over five runs and includes error bars.

- **Real-world compositional benchmark (dojw).**
  Reviewer dojw asked for a more realistic synthetic benchmark. We clarify that **TWOHOPFACT** is *already* a real-world compositional benchmark: it is constructed from factual triples extracted from real knowledge bases (e.g., *novel–author–birthplace*), and the two-hop queries involve real entities and relations rather than arbitrary synthetic symbols.

- **Multi-step reasoning (rRJu).**
  Reviewer rRJu requested results on longer reasoning chains. In Section 4.5 of the revised manuscript, we added new experiments and a detailed analysis of **three-hop** reasoning. We observe that the behavioral patterns in the three-hop setting closely mirror those in the two-hop setting. This consistency suggests that the mechanisms we identify are not specific to two-hop tasks, but extend to longer reasoning chains.

**Reference**

[1] Boshi Wang, Xiang Yue, Yu Su, and Huan Sun. *Grokking of implicit reasoning in transformers: A mechanistic journey to the edge of generalization.* NeurIPS 2024.

We thank the reviewers and area chairs again for their careful reading and constructive feedback.

---

### Meta-Review · Area_Chair_NkCw · 2025-12-19

**Summary:**

The paper considers two hop reasoning tasks, and aims to improve the performance of LLMs on such tasks. It introduces an identity bridge: a relation pair relating an entity to itself. It shows that including identity bridges in the training set helps in 2 hop reasoning performance of LLMs. The paper also theoretically examines the scope of this improvement.

The reviewers raised several concerns (discussed more below). Due to these, the paper is short of the bar.

My opinion is that the paper can aim to either show a more direct relevance of the results to real transformers, or further examine the conceptual implications. The fact that the identity bridge helps points to an interesting phenomenon which the paper explores to some extent with the theory, but it could possibly have broader implications for OOD generalization for reasoning tasks. There may be a bigger and more interesting story here. It is also possible that significant rewrites (such as what authors did in the revision period) help.

**Reviewer Concerns:**

Reviewers raised several concerns. Several of them had concerns around the theoretical assumptions. They also had concerns regarding application to real transformers, and the significance of the findings on real data. I believe that addressing all of these thoroughly is outside the scope of a revision, and requires resubmission.

There was also a concern regarding CoT, though I think that is okay, it is interesting to examine the behavior without CoT too.

**Reviewer Scores:**

Scores may have changed a bit, but significant enough changes are unlikely.

---

### Decision · Program_Chairs · 2026-01-26

Reject